# Characterizing pre-transplant and post-transplant kidney rejection risk by B cell immune repertoire sequencing

Silvia Pineda [1,2], Tara K. Sigdel[2], Juliane M. Liberto[2], Flavio Vincenti[2], Marina Sirota[1,3,4] & Minnie M. Sarwal[2,4]

Studying immune repertoire in the context of organ transplant provides important information on how adaptive immunity may contribute and modulate graft rejection. Here we characterize the peripheral blood immune repertoire of individuals before and after kidney transplant using B cell receptor sequencing in a longitudinal clinical study. Individuals who develop rejection after transplantation have a more diverse immune repertoire before transplant, suggesting a predisposition for post-transplant rejection risk. Additionally, over 2 years of follow-up, patients who develop rejection demonstrate a specific set of expanded clones that persist after the rejection. While there is an overall reduction of peripheral B cell diversity, likely due to increased general immunosuppression exposure in this cohort, the detection of specific IGHV gene usage across all rejecting patients supports that a common pool of immunogenic antigens may drive post-transplant rejection. Our findings may have clinical implications for the prediction and clinical management of kidney transplant rejection.

[1] Bakar Computational Health Sciences Institute, University of California, San Francisco (UCSF), 550 16th Street, San Francisco, CA 94158, USA. [2] Division of Transplant Surgery, Department of Surgery, University of California, San Francisco (UCSF), 505 Parnassus Ave, San Francisco, CA 94143, USA. [3] Department of Pediatrics, University of California, San Francisco (UCSF), 550 16th Street, San Francisco, CA 94158, USA. [4] These authors are contributed equally: Marina Sirota, Minnie M. Sarwal. Correspondence and requests for materials should be addressed to S.P. (email: silvia.pinedasanjuan@ucsf.edu) or to M.S. (email: marina.sirota@ucsf.edu) or to M.M.S. (email: minnie.sarwal@ucsf.edu)

K idney transplantation is the preferred treatment of end-stage renal disease (ESRD) and chronic kidney disease. Even though there have been significant improvements in technologies and tissue matching based on histocompatibility testing for donor/recipient human leukocyte antigens (HLA)[1], predicting graft outcome is still an unsolved problem. Unmeasured tissue mismatching at other minor, non-HLA loci[2,3] can also drive alloimmune injury with acute and chronic rejection, resulting in poor long term graft outcomes[4]. Moreover, around 50% of kidney allografts, with no major HLA mismatches, are still lost within 10 years of transplantation[5]. We have previously hypothesized that non-HLA loci, may influence the immune response of the recipient against his kidney donor graft[6]. It is clear that more research needs to be done to better understand and predict the recipients' risk of rejection to substantially improve long term patient and graft outcomes. Nevertheless, the diversity of the immune response to various immunogenic epitopes is as yet, poorly understood. The role of T cells in organ transplant rejection has been demonstrated[7], but there is increasing appreciation of the additional role of B cells and antibodies in triggering this process[8]. In this regard, B-cell receptor sequencing (BCRSeq) is a promising high-throughput technique[9] that allows the sequencing of millions of Immunoglobulin (Ig) regions in parallel to study the immune response. The key feature of B cells is their enormous diversity. Each individual is capable of producing $>10^{13}$ different antibodies[10], which enables them to recognize a vast array of foreign antigens. Human BCRs or Ig consist of two identical heavy chains (IgH) formed by five isotypes: IgM, IgD, IgA, IgE, and IgG and two light chains. The intact antibody contains a variable and a constant domain. Antigen binding occurs in the variable domain, which is generated by recombination of a set of variable (V), diversity (D) and joining (J) gene segments forming the B-cell immune repertoire, and its diversity is mainly concentrated in the complementary determining region 3 (CDR3). During the process of affinity maturation, somatic hypermutation (SHM) occurs in the variable region. A potent adaptive immune response is reliant upon the expansion of B-cell clones and a process termed affinity maturation, during which somatic mutations are introduced into the Ig gene rearrangements and B cells with higher affinity for a given antigen are selected.

The study of the immune repertoire in organ transplant is crucial to understand what triggers and sustains the rejection process and how it may eventually accelerate the path toward graft failure. With the advances of next generation sequencing and robust computational approaches, we can study the VDJ region in fine detail[11,12]. To date, T-cell immune repertoire analysis in kidney transplant has been carried out in very limited numbers of patients[13–15], and even though BCRSeq has been applied to other diseases and human immune responses, such as multiple sclerosis[16], influenza vaccine[17] or immunodeficiency disorders[18], there is a lack of studies in transplant rejection. In kidney transplant, BCRSeq has been only carried out in the context of tolerance[19], HLA sensitized kidney transplant candidates undergoing desensitization therapy[20] and B-cell infiltration comparing clonal expansion in blood and graft[21]. Another application on BCRSeq in transplant was previously published in a small cohort of 12 heart transplant recipients[22].

Recognizing the importance of the humoral arm of immune response in late transplant rejection and chronic allograft failure, we characterize the peripheral blood immune repertoire using BCRSeq in a prospective, longitudinal study. We find that the immune repertoire diversity before transplantation is higher in those individuals who reject the kidney showing also expansion of certain clones and IGHV genes along the 24 months of follow-up. These results may help predict the rejection risk before

engraftment, and may have clinical implications in the detection of particular antigens driving rejection.

## Results

**Study subjects.** We performed BCRSeq in 83 peripheral blood samples from 27 unique patients and executed the analytical pipeline shown in Fig. 1. Three clinical phenotype groups, defined by blinded central pathology reads of serial allograft biopsies scored by Banff criteria[23,24] and the chronic allograft damage index (CADI) score were considered in this study: Non-progressors (NP; $n = 10$) had low non-incremental CADI score without acute rejection, progressors with no rejection (PNR; $n = 10$) had incremental CADI score over 2 years without rejection, and progressors with rejection (PR; $n = 7$) had incremental high CADI scores over 2 years with rejection episodes. Demographics, causes of kidney failure and immunosuppression usage is provided in Table 1. It is important to highlight some characteristics about these patients. The parent study that these patients were enrolled from had an overall low rate (17%) of biopsy confirmed acute rejection (mean{min,max} = 12 {6,24} months rejection time). These patients were all at low immunologic risk for rejection (peak panel reactive antibody sensitization status < 20%), and also had low rates of generation of donor-specific antibody (DSA) and in fact only two of the rejection phenotype patients included in the analysis had DSA. The generation of DSA to HLA, and MICA were measured in all serial sera over the course of the study[25]. National experience with similar immunosuppressive protocols in similar patient cohorts have confirmed similar good clinical outcomes and low rejection rates[26–28].

**B-cell immune repertoire sequencing.** BCRSeq was done on genomic DNA (gDNA) samples extracted from blood clots on 81 samples from 27 kidney transplant recipients at three time points (0, 6, 24 months). Sequencing obtained a total number of 327,703 reads (mean 4045/sample) after quality control (see methods section). For validation of the results and further evaluation of each isotype, we additionally extracted RNA from matched PBMC that were available for 55 samples, collected at the same time as the blood clot, and performed complementary DNA (cDNA) sequencing at greater depth obtaining 1,773,330 reads for IgD (mean 31,667/sample), 1,708,227 reads for IgM (mean 30,504/sample), 973,444 reads for IgA (mean 17,383/sample), 139,7345 reads for IgG (mean 24,953/sample), and 29,000 reads for IgE (mean 5178/sample) (Supplementary Fig. 1). Libraries for each isotype were amplified separately and then pooled for sequencing; therefore, comparative cross isotype analysis is not feasible.

As shown in Fig. 1b, we defined a clone as a group of cells descended from a common ancestor molecule that have the same IGHV and IGHJ segment, same CDR3 length, and 90% nucleotide identity between CDR3s as previously defined in studies of adaptive B-cell responses[18]. This definition allows to study diversity, shared or common clones, and clonal expansion in the context of alloimmunity in kidney transplant. The number of unique clones per individual at each time point is shown as a barplot in supplementary material (Supplementary Fig. 1). For stringency of data analysis, we discounted samples with <100 clones (69 samples from gDNA and 55 samples from cDNA were left for further study. IgE isotype was discarded completely due to a very small number of reads). In addition, we filtered out patient 8 in the NP group from subsequent analysis, as we recognized after the run, that he had developed EBV + post transplant lymphoproliferative disease (PTLD) at 2.2 years post kidney transplant, characterized by proliferation of Epstein Barr

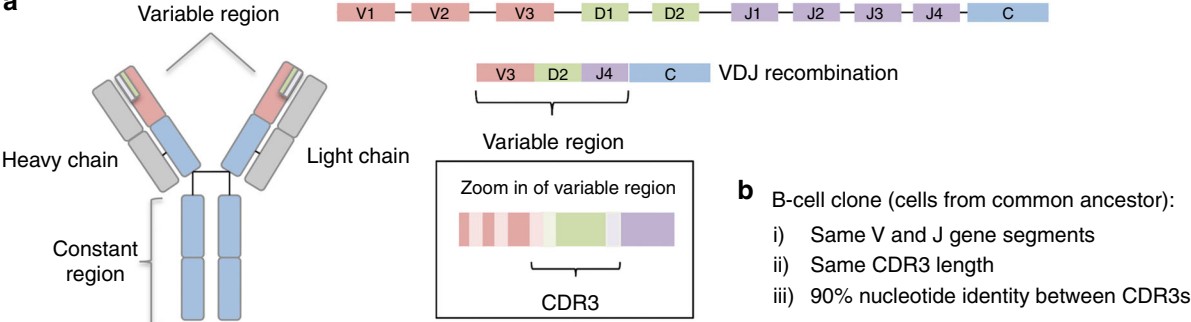

**a**

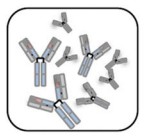

**b**  B-cell clone (cells from common ancestor):

i)    Same V and J gene segments

ii)   Same CDR3 length

iii)  90% nucleotide identity between CDR3s

**c**

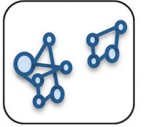

B-cell sequencing (83 samples):
- gDNA: 81 samples on 27 recipients at 0, 6, 24 months
- cDNA: 56 samples on 27 recipients at 6, 24 months

Diversity analysis (richness and Shannon's entropy) to find association with clinical outcome (NP, PNR, PR):
- At each time point: linear models
- Longitudinally: linear mixed effect models (interaction time*clinical outcome)

B-cell repertoires represented by networks:
- Network plot:
    Vertex: unique B cell sequences
    Edges: B cell sequences belonging to the same clone
- Network analysis: Gini index applied to vertex and cluster distribution

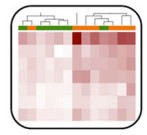

Clonal and IGHV gene usage analysis to find association with clinical outcome:
- Clonal analysis:
    Univariate (Fisher's exact test)
    Persistence clones (total number and expansion)
- IGHV gene usage analysis
    At each time point: linear models

**Fig. 1** Overall study pipeline. **a** Schematic representation of antibody structure and the process of VDJ recombination responsible for the diversity produced in the immune repertoire. **b** B-cell clone defined as cells from a common ancestor and **c** analytical pipeline of the study: B-cell sequencing for gDNA and cDNA, diversity analysis considering the number of clones (richness) and the frequency of each clone (Shannon entropy), network analysis and clonal and IGHV gene analysis

| Table 1 Subjects characteristics included in the study | | | | | | |
|---|---|---|---|---|---|---|
| | | Total | NP | PNR | PR | *P*-value |
| Total subjects | | 27 | 10 | 10 | 7 | |
| Age donor | Mean [min–max] | 30 [15–47] | 26 [15–47] | 33 [17–44] | 31 [20–42] | 0.2 |
| Age recipient | Mean [min–max] | 12 [1–19] | 13 [8–19] | 10 [3–19] | 11 [1–17] | 0.3 |
| Gender donor | Male/female | 14/13 | 7/3 | 5/5 | 2/5 | 0.2 |
| Gender recipient | Male/female | 16/11 | 5/5 | 6/4 | 5/2 | 0.7 |
| Donor source | Cadaver | 13 | 6 | 4 | 3 | |
| | Living/related | 14 | 4 | 6 | 4 | 0.6 |
| Race recipient | Caucasian | 12 | 2 | 6 | 4 | |
| | AA | 6 | 4 | 1 | 1 | 0.2 |
| | Asian | 1 | 0 | 0 | 1 | |
| | Other | 8 | 4 | 3 | 1 | |
| Immunosuppression | Steroid-based | 13 | 6 | 4 | 3 | |
| | Steroid-free | 14 | 4 | 6 | 4 | 0.6 |
| HLA mismatch | Mean [min–max] | 4 [1–6] | 4 [3–6] | 4 [1–6] | 4 [2–6] | 0.8 |
| Cause of ESRD | Non-immune structural mediated | 4 | 2 | 2 | 0 | 0.6 |
| | Other/unknown | 13 | 5 | 5 | 3 | |
| | Reflux | 10 | 3 | 3 | 4 | |
| Rejection time (months) | Mean [min–max] | | | | 12.5 [6.0, 24.0] | |
| DSA | | 2 | 0 | 1 | 1 | - |

There were no significant differences for any of the donor and recipient characteristics by clinical outcome, including HLA mismatch and end-stage renal disease (ESRD) that was classified in three main groups: Non-immune structural mediated (polycystic kidney disease and aplastic/hypoplastic/dysplastic kidneys), Reflux (obstructive uropathy, pyelonephritis/interstitial nephritis and reflux nephropathy) and other/unknown (focal segmental glomerulosclerosis, cystinosis, hemolytic uremic syndrome, cortical necrosis, other and unknown). Class I DSA was developed in one individual in the PNR and also was found at time 0 in another individual in the PR group

virus (EBV) infected B cells. We observed an increased number of clones at time 6 in this patient, with lower clonal diversity before kidney transplant and at 24 months after kidney transplant. Since libraries were amplified from an invariant amount of template, a higher fraction of B cells in the blood may have resulted in a larger number of clonotypes represented in the sequenced products. Because of the distinct and unique pathologic process in this patient, this sample was excluded from future analysis.

**Pre-transplant B-cell diversity is associated with rejection**. As shown in Fig. 1c, we examined the B-cell immune repertoire diversity considering species richness (number of unique clones) and Shannon entropy (equation (1) from methods) across time points and clinical outcomes using a linear regression model considering the number of clones or entropy as a dependent variable and clinical outcome or SHM as an independent factor variable. The repertoire before kidney transplant in PR was significantly more diverse than in the NP (richness: P-value = 0.005, entropy: P-value = 0.01) with the same trend persisting at 6 months after kidney transplant (richness: P-value = 0.02, entropy: P-value = 0.02), and with no distinguishable group differences at 2 years post transplant (Fig. 2a and Supplementary Fig. 2). cDNA sequencing data post transplant showed the same trend in greater repertoire diversity at 6 months after transplant, predominantly for the IgD isotypes (richness: P-value = 0.02, entropy: P-value = 0.03) (Fig. 2b and Supplementary Fig. 2). There was no confounding effect on the data from various demographic and clinical variables, such as recipient age, gender, race, donor source, type of immunosuppression, HLA mismatch, and cause of renal failure. Since the B-cell response of a 1-year-old (the minimum age in the data) and 19-years-old (the maximum age in the data) might be very different, we performed a sensitivity analysis excluding these two patients and the results remained significant (NP vs. PR at time 0: richness: P-value = 0.01, entropy: P-value = 0.03). In another measure of immune repertoire diversity, we evaluated the SHM, defined as the frequency of mutations in each V gene segment, and found a trend for higher number of SHM for PR before transplant, and trend for higher SHM in the IgD isotype in PR at 6 months post transplant (P-value = 0.06).

**B-cell diversity changes across time by clinical outcome**. To find whether the immune repertoire diversity changes across time by the clinical outcome, we modeled the longitudinal data using linear-mixed effect models considering the interaction between clinical outcome and time. We found that NP and PR behaved differently across time after transplant showing an increase in diversity in NP and a decrease in diversity in PR, while for PNR the diversity remained invariable across time. This was observed for gDNA (richness: P-value = 0.007, entropy: P-value = 0.001, Fig. 3a) and all isotypes for cDNA, with the most significant differences in entropy being for IgM and IgD isotypes (IgA: P-value = 0.07, IgD: P-value = 0.02, IgG: P-value = 0.05, IgM: P-value = 0.04, Fig. 3b). The plots for richness with corresponding P-values are shown in Supplementary Fig. 3. As observed in the plots, one individual has an extra sample at time 32 months, which might skew the results, thus a sensitivity analysis was performed excluding this sample from the analysis and observing similar results except for the IgG and IgM isotypes where the significance is lost (gDNA-entropy: P-value = 0.004. cDNA-entropy: IgA: P-value = 0.1, IgD: P-value = 0.05, IgG: P-value = 0.08, IgM: P-value = 0.1).

Diversity measures may be affected by biological and technical sampling. The diversity of a sample can differ markedly from the overall diversity in a repertoire since only a fraction of billions of cells are represented, which is known as missing species problem. Technical sampling may exist because each sample may vary on sequencing depth and some degree of experimental errors. To deal with the missing species problem, we used Recon (reconstruction of estimated clones from observed numbers) tool[29], which estimates the overall clone size distribution. Recon outputs accurate and robust estimates of a set of diversity measures, including richness and entropy allowing robust comparisons of diversity between individuals. To deal with the sequencing depth and some degree of experimental errors, we performed a downsampling strategy; a very well used strategy in immune repertoire analysis[11,17,30]. In the Recon analysis (Supplementary Fig. S4: A–E), both richness and diversity were replicated for the gDNA data. In the cDNA data, time 6 IgD isotype shows the trend but did not reach significance although the longitudinal results for IgA, IgD, and IgG isotypes were replicated. In the downsampling analysis (Supplementary Fig. S4: F–J), everything is replicated but the time 0 for gDNA data. This might be a consequence of the restriction that downsampling

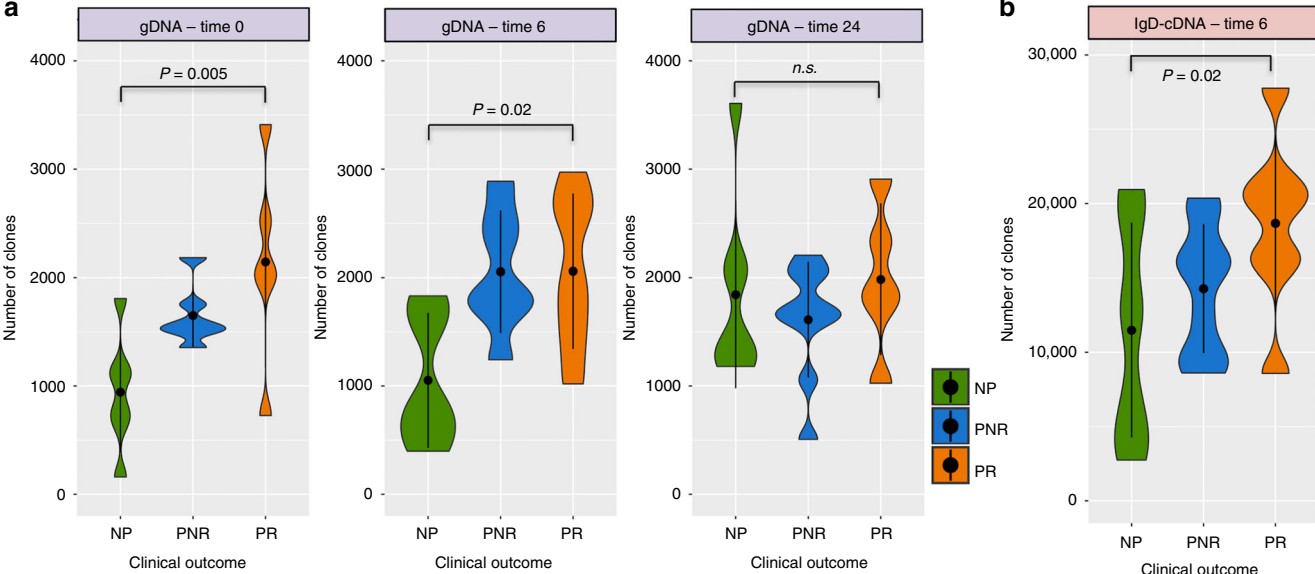

**Fig. 2** Violin plots showing the number of clones (richness) across the three clinical outcomes. **a** Number of clones at time 0, 6, and 24 from gDNA samples. **b** Number of clones at time 6 for the IgD isotype from cDNA samples. The P-values are obtained from the adjustment of a linear regression model considering the number of clones as a dependent variable and clinical outcome as an independent factor variable (n = 27 samples). Violin plots represent the probability density of the data at each value. The dot marker represents the median value with the interquartile range. Source data is provided as Source Data File

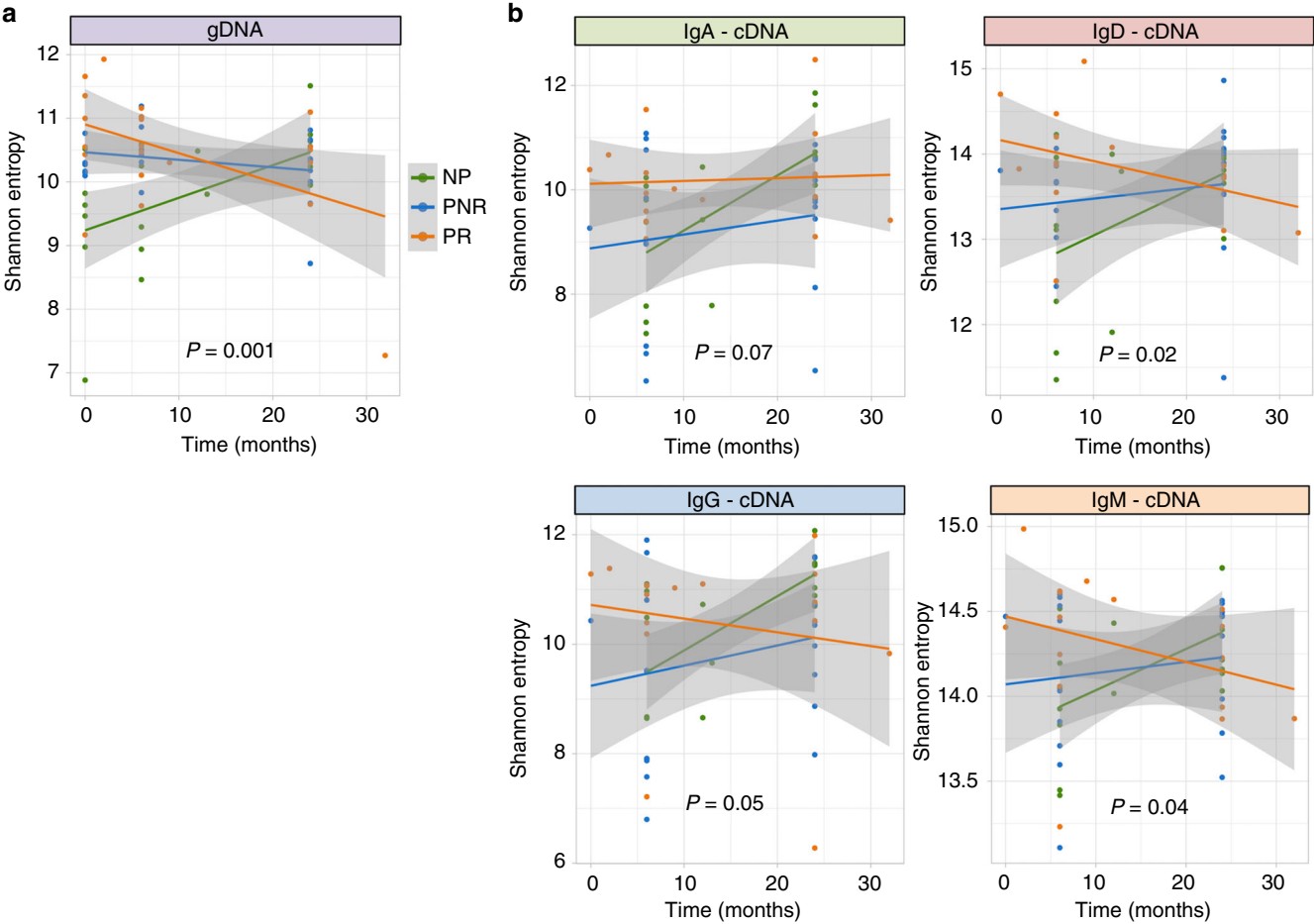

**Fig. 3** Longitudinal data plotted with the fitted line for each clinical outcome. **a** Diversity measured by Shannon entropy represented across time points by the three clinical outcomes (NP, PNR, PR) in gDNA. **b** Diversity measured by Shannon entropy represented across time points by the three clinical outcomes (NP, PNR, PR) by isotypes in cDNA. The *P*-values correspond to the interaction term defined by time × clinical outcome adjusting a linear-mixed effect model (*n* = 27 samples). Each dot represents the entropy per sample along the time with its fitted line and confidence interval. Source data is provided as Source Data File

strategies implied, especially on the gDNA data where the number of sequences is much lower than for the cDNA data. In gDNA, we restricted the analysis to individuals with at least 1000 clones in order to conserve enough sequences. Downsampling was done to a minimum of 1062 clones in comparison to 62,173 clones in cDNA. Despite this limitation, we observed the trend for time 0, conserve the significance for the longitudinal analysis in gDNA and replicated all the associations in cDNA. In both analyses (recon and downsampling), we replicated the results, despite some limitations highlighted here, showing the validity of the previously reported results.

**B-cell networks show differences in clonal expansion**. The B-cell repertoire can be naturally represented as a network based on sequence diversity[31]. In our data, we developed a visual network for each sample (Fig. 4 and Supplementary Figs. 5–7) where each vertex represented a unique BCR, and the number of identical BCRs based on their nucleotide sequences defined the vertex size. An edge exists between vertices when they belong to the same clone, so clusters of B cells can be shown as groups of interconnected vertices forming a clone. To quantify the network, we used the Gini index, which is an unevenness measure that was applied to the vertex and cluster distributions. When applied to vertex size, *Gini(V)*, the overall clonal nature is represented. If

*Gini(V)* is closer to 1, vertices are unequal showing expansion of some of them, and closer to 0 otherwise. When applied to cluster size, *Gini(C)*, clonal dominance is represented. If closer to 1, clusters are unequal and therefore represent dominant clones, if closer to 0, all clusters are of equal size.

In Fig. 4 we show an example of the marked visual and quantitative differences between representative B-cell repertoires from each of the three clinical outcome groups, across the three different time points (pre-transplant, and post transplant 6 and 24 months). In the PR repertoire, there is an abundance of B-cell sequences of forming more and larger clusters of clones in comparison with NP while PNR is located in between. Across time, the PR group shows a decrease in the number of BCR and unique clones in comparison with NP. The detailed B-cell networks of every individual in the study are provided in Supplementary Figs. 5–7. It is interesting to observe the very diverse pattern of the B-cell network in the individual who developed PTLD in the NP group (Supplementary Fig. 5), with no B-cell expansion before transplant, followed by an increase of B cells after 6 months and a clear clonal expansion accompanied with a decrease in diversity at 24 months post transplant.

On further evaluation of the Gini Index measure (Fig. 5), the PR group consistently showed significantly higher measures for both the vertex and the cluster over the NP group, suggesting that the PR group patients had higher clonal expansion at baseline,

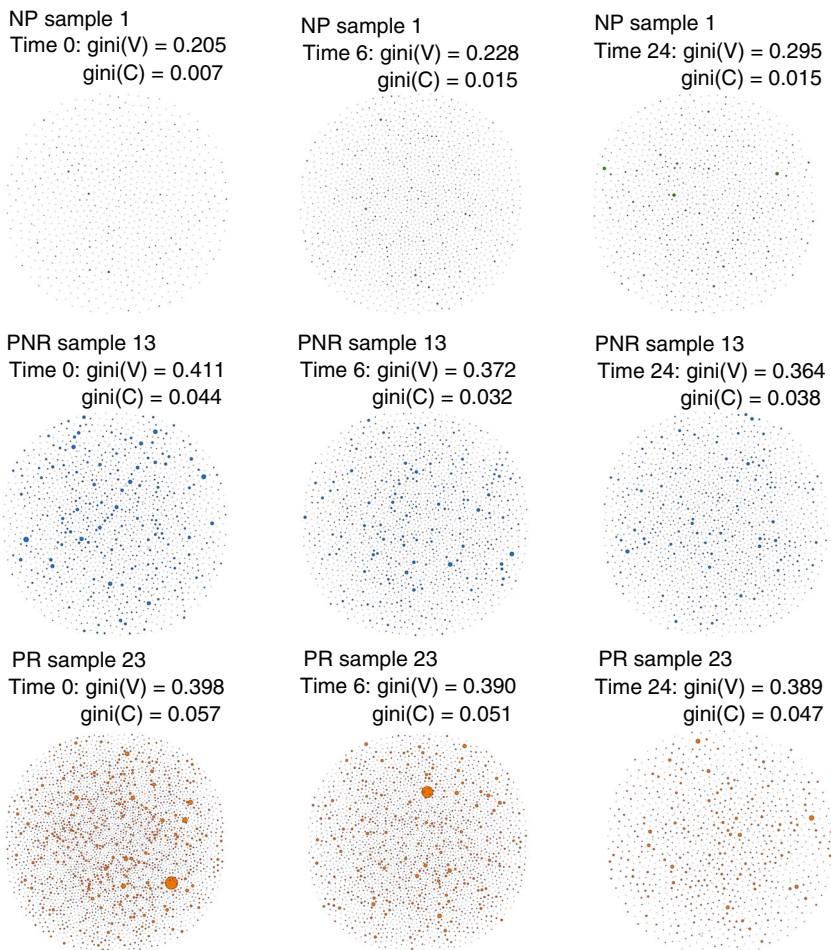

**Fig. 4** B-cell repertoire networks from three individuals representing the three clinical outcomes across time points. Each vertex represents a unique BCR being the vertex size defined by the number of identical BCRs considering the nucleotide sequences. An edge exists between vertices when they belong to the same clone as defined before, so clusters are groups of interconnected vertices forming a clone. Each sample shows the gini index obtained for the vertex size (*Gini(V)*) and cluster size (*Gini(C)*). BCR reflects the total B-cell receptors for that specific sample and clones reflect the total number of unique clones. Source Data is provided as Source Data File

and further post transplant expansion of a subset of dominant clones (*P*-value [linear regression] < 0.05, Fig. 5). The PNR group falls between NP and PR as previously shown, although this time is significantly different over the NP group at time 24 (*P*-value [linear regression] < 0.05). Although the data for Fig. 5 was generated from the gDNA data, the IgM isotype for cDNA showed the same differences (*P*-value [linear regression] = 0.05) (Supplementary Fig. 8).

**Certain clones and *IGHV* genes are implicated in rejection.** Although our primary focus was to characterize the B-cell repertoire by clinical outcome groups, providing a global picture of the immune response pre- and post transplant, our data also enabled us to perform Ig sequence-specific analysis at the clone and the *IGHV* gene level.

For clonal analysis, we assessed the association of the presence or absence of each particular clone (118,223 total clones) with the clinical outcome (PR, PNR, NR) at each time point. Applying Fisher's exact test, we found 8, 4, and 21 clones nominally associated with clinical outcomes at each of 0, 6, and 24 months, respectively (Supplementary Table 1). While none passed multiple testing correction, mainly because of a lack of power since we have a limited sample size in an analysis with thousands of parameters (clones), we could observe that the few clones that

approached significance (*P*-value < 0.05) were shared across patients only in PR group and enriched at 24 months post transplant.

We also considered whether there were some clones that persisted over sampling time, more than others, within each clinical outcome (Supplementary Fig. 9). We account for two different measures: (1) number of persisting clones and (2) clonal expansion. We observed that the clones that were persistent in the PR were significantly more expanded (*P*-value [linear regression] = 0.01, PR vs. NP) and showed a trend of a higher number of persistent clones (*P*-value [linear regression] = 0.09). We further examined whether these persistent clones were also shared across different individuals within each clinical outcome. From the 263 persistent clones detected, 23 were shared across individuals. Five were shared within the same PNR and six within the PR group and no clones were shared between the NP group. In total, there were 12 shared clones that were common across both groups of patients with progressive chronic transplant injury and fibrosis over time (PR and PNR). The list of clones shared across individuals is provided in supplementary material (Supplementary Table 2).

We next performed *IGHV* gene analysis, looking at *IGHV* gene usage per sample, defined as the number of times each *IGHV* gene has been used, normalized by the number of clones (to avoid sampling bias of certain *IGHV* genes), filtering out low-expressed

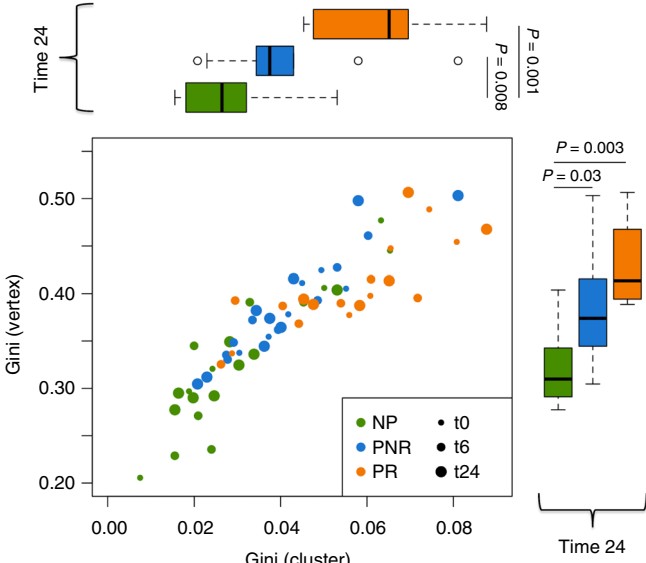

**Fig. 5** Vertex Gini Index plotted against Cluster Gini Index. The scatter plot represents each sample at time 0, 6, and 24. Boxplots shows the *Gini(V)* and *Gini(C)* differences at time 24. The *P*-values are obtained from the adjustment of a linear regression model considering the *Gini(V)* and *Gini(C)* as a dependent variable and clinical outcome as an independent factor variable for each time point (*n* = 27 samples). In the boxplot only time 24 is shown but time 0 and 6 where also significant for NP vs. PR: time 0: *P* (*Gini (V)*) = 0.1, *P* (*Gini(C)*) = 0.05; time 6 *P* (*Gini(V)*) = 0.02, *P* (*Gini(C)*) = 0.01; time 24: *P* (*Gini(V)*) = 0.003, *P* (*Gini(C)*) = 0.01. The band inside the box represents the median value, the box define the interquartile range (IQR) and whiskers define the first and third quartile ± 1.5 × IQR. Source data is provided as Source Data File

genes (*IGHV* gene usage > 0.05 in at least 10% of the samples), and applying a linear regression model to find those genes that were associated with each clinical outcome, at each time point. From the 27 *IGHV* genes that passed the low-expression filter, we found significant genes between the PR and NP group (*P*-value < 0.05) with three genes at time 0, 7 at time 6 and 16 at time 24 (Fig. 6a–c). From these genes, 1 (*IGHV3-11*) at time 0, 5 genes (*IGHV3-7, IGHV3-15, IGHV3-21, IGHV3-23, IGHV4-39*) at time 6 and 16 genes (*IGHV1-8, IGHV1-18, IGHV1-46, IGHV2-5, IGHV3-7, IGHV3-11, IGHV3-15, IGHV3-23, IGHV3-30, IGHV3-33, IGHV3-48, IGHV3-74, IGHV4-39, IGHV4-59, IGHV4-61, IGHV5-51*) at time 24 passed False Discovery Rate (FDR) multiple testing correction. Interestingly, we found that *IGHV3-23* was the most significant and abundant gene across all three time points in the NP vs. PR comparison (time 0: *P*-value = 0.04, time 6: *P*-value = 0.003, time 24: *P*-value = 0.02) (Fig. 6d). In addition, we evaluated whether the *IGHV3-23* sequences were over represented among the shared sequences from the previous clonal analysis. We found, using an enrichment analysis with Fisher's exact test, that the *IGHV3-23* sequences were significantly over represented in both, the persistent clones shared among individuals (Supplementary Table 2) (*P*-value < 2.2 × 10$^{-16}$), and the clones associated with clinical outcome at time 24 (Supplementary Table 1) (*P*-value < 2.2 × 10$^{-16}$). We observed that individual 9 in the NR group, who had been found to share some persistent clones with the progressors in the previous analyses, classified with the PR group at all time points. There was no confounding effect on the data from various demographic and clinical variables, such as recipient age, gender, race, donor source, type of immunosuppression, HLA mismatch, and cause of renal failure. In addition, this gene was also found to be

significant in both the IgM (*P*-value [linear regression] = 0.008) and IgD (*P*-value [linear regression] = 0.05) isotypes based on cDNA at 24 months post transplant, in concordance with the previous results showing consistency with these two isotypes being most enriched in the PR group. Only three other genes were found significant in the cDNA analysis, and all were at 24 months post transplant in IgD and IgM isotypes (*IGHV3-15* and *IGHV4-61* in IgD and *IGHV4-39* in IgM).

## Discussion

Diversity is an essential characteristic of the immune system, and in healthy humans, is critical against pathogens to deal with diseases. In organ transplant, deliberate therapeutic manipulation is instituted clinically to allow the foreign organ to be actively ignored or accepted by the own patient's immune system so as to not mount an alloimmune response, leading to rejection. B cells are an important component of this process and have been shown by our group and others, to be pivotal in both antigen presentation[32,33] and alloantibody production[34,35]. In this work, we have used high-throughput B-cell sequencing to better understand the diversity and clonality of B cells in the kidney transplant recipient's circulation, both before engraftment and after 24 months of follow-up, with longitudinal assessment of the B-cell immune repertoire. Overall, our analysis shows a higher B-cell diversity before engraftment, differences longitudinally with a decrease in diversity accompanied by clonal expansion and an increase in certain *IGHV* gene usage among those who go on to reject the grafts.

A key unmet clinical need in organ transplant is the lack of noninvasive, sensitive, and accurate prediction of transplant injury and poor outcomes. This task is complicated by the fact that there are diverse factors that influence graft survival[36]. In this study, we found that stable individuals had a reduced diversity of the B-cell immune repertoire before transplant in comparison with those who rejected the organ. The next step should be to demonstrate the predictive value of B-cell repertoire diversity, providing potential better biomarkers for prediction of rejection before engraftment, and the possibility of being implemented in clinical care and immunosuppression choices before and after kidney transplant. If this feature is the result of any factor contributing to the decrease of diversity in stable individuals, such as environmental or genetic factors, we could not only predict the rejection but also prevent it. Indeed, very recent findings showed that the variation in the immune system is driven by environmental and genetic factors[37,38]. In our study, we could not find any association with any demographic and other clinical patient characteristics (age, gender, race, immunosuppression, organ source, HLA mismatches, and ESRD). In this regard, we need new analysis to corroborate whether specific factors affect the B-cell repertoire diversity and transplant outcomes.

Another interesting finding in our study shows that the immune repertoire behaves differently across time depending on the clinical outcome group. For those who show post transplant rejection of the organ, the B-cell diversity is initially higher and then decreases over time, whereas the reverse diversity trend is seen in patients who do not develop rejection or chronic graft injury. Even though all individuals received the same immunosuppression load after transplant, some possible explanation for the decreased diversity in the patients who develop rejection, may relate to the fact that these patients receive a temporarily greatly increased load of immunosuppression for the treatment of rejection and then are kept on higher baseline. Though this could explain the reduction in B-cell diversity over time, and explain the difference at 24 months post transplant, this is unlikely to be the cause, as the reduced diversity in the PR group is

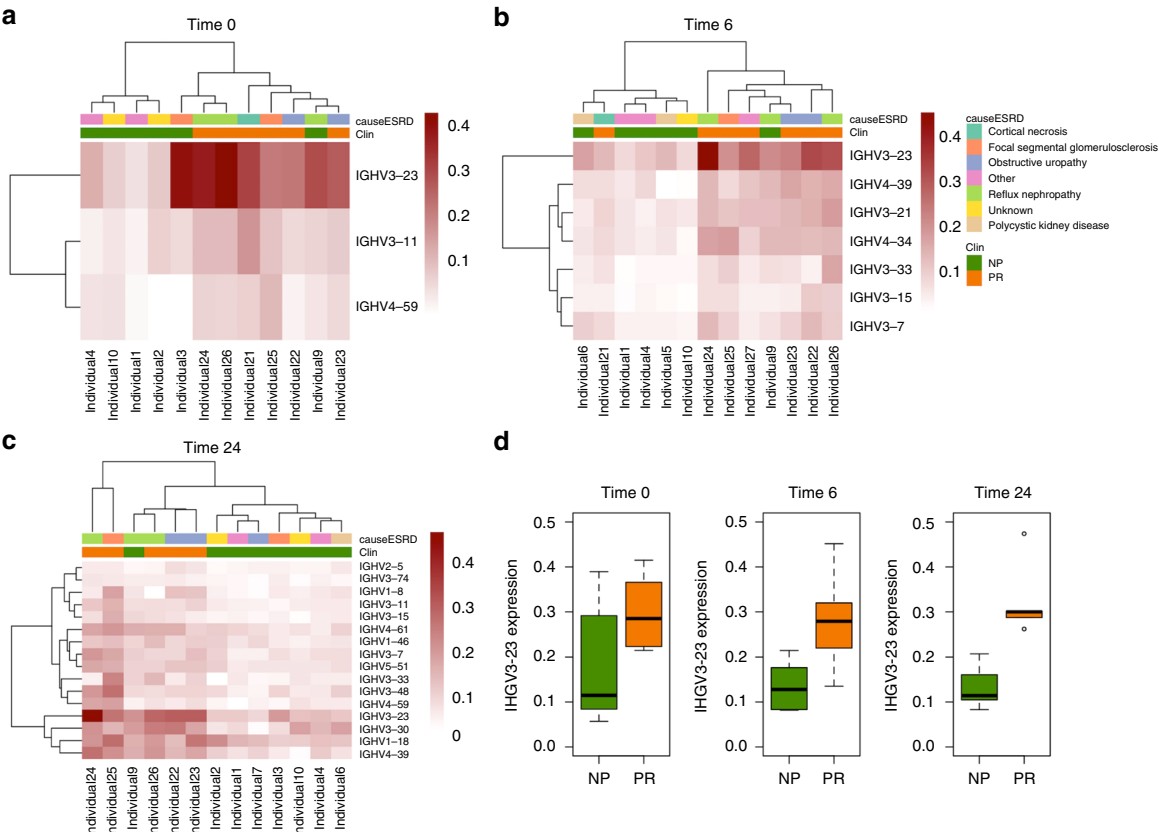

**Fig. 6** Heatmap and boxplot for the *IGHV* genes usage analysis. Heatmap showing the *IGHV* genes selected as nominally significant (*P*-value < 0.05) at time 0 (**a**), 6 (**b**), and 24 (**c**) for NP vs. PR. The red color scale represents the expression of each gene across samples. The bands on the top shows the clinical outcome and the cause of ESRD. Boxplot showing *IGHV*3-23 expression across time points for NP vs. PR (time 0: *P*-value = 0.04, time 6: *P*-value = 0.003, time 24: *P*-value = 0.02) (**d**). The band inside the box represents the median value, the box define the interquartile range (IQR) and whiskers define the first and third quartile ± 1.5 × IQR. *P*-values are obtained from the adjustment of a linear regression model considering the V genes as a dependent variable and clinical outcome as an independent variable. (*n* = 12 samples). Source data is provided as Source Data File

also seen at 6 months post transplant, at a time when patients have not yet developed rejection, suggesting that other active processes are likely to be involved. The observation of selective clonal expansion with more dominant clones only in patients who develop both rejection and/or progressive chronic kidney transplant injury, suggests a biologically relevant, alloantigen driven selection, persistence and expansion of certain clones over time, which accounts for the overall reduction in temporal diversity. Even though we can hypothesize that the expansion of these clones is likely linked to alloimmnue injury to the graft, direct evidence that the expansion of these clones is alloimmune requires additional in vitro and animal studies.

Acute rejection remains the strongest negative factor for long term graft survival despite the rapid improvements in immuno-suppression therapies[39,40]. This study also finds an association of some *IGHV* genes usage with rejection, although the causal link should be explored in future studies. The *IGHV*3-23 is the most interesting gene in the gDNA analysis as it is significantly higher used in patients who develop rejection across all measured time points and it is over represented in the persistent clones shared among individuals and the clones enriched among rejected patients at 24 months post transplant. This gene is also validated at 24 months post transplant in both the IgD and IgM isotypes in the cDNA analysis. *IGHV*3-23 has been extensively associated with bad prognosis in chronic lymphocytic leukemia[41] and it has been shown that the vast majority of *IGHV*3-23 sequences retained the capacity to mediate superantigen interactions[42]. B-cell superantigens are produced by viruses and bacteria, known

to bind to immunoglobulins outside the conventional antigen binding sites[43]. Encounters of B cells with a superantigen have been shown to induce proliferation, activation, migration and deletion[44]. In the transplanted kidney, there is the potential for continuous exposure to virus and bacterial antigens, relating either to an etiology of primary vesicoureteric reflux as the cause of ESRD or secondary urinary reflux after reimplantation of the transplant ureter, or reflux following urinary tract infection, the risk of which are increased with exposure to chronic immuno-suppression after transplantation. We adjusted the analyses results for cause of renal failure, among other clinical and demographic factors; the results previously discussed remain significant although we realized that the NP misclassified patient in the analysis (Fig. 6) had a reflux disease. Cheng et al. have also previously found that clones in biopsy tissue of kidney transplant recipients infiltrated by B cells had a predominance of the *IGHV*3-23 gene among others[45]. Grover et al.[46] demonstrated that the antibodies for this particular gene were not recognizing donor HLA antigens but rather were specific for *E. coli* and Modena et al.[47] showed that the burden of the number of bacteria in urine were much higher in patients with interstitial fibrosis and tubular atrophy than those with grafts that were functioning well. We extend these findings demonstrating that the use of *IGHV*3-23 gene is significantly higher in multiple rejecting patients in peripheral blood and may implicate particular com-mon antigens in driving rejection. In the future, the expression of *IGHV*3-23 could be potentially used to monitor the immune response towards the transplanted kidney.

Extending our study using cDNA sequencing, we could replicate the majority of our results in two of the isotypes (IgM and IgD), showing lower diversity pre-transplant in the NP, being the two most significant in the longitudinal analysis with more expansion and dominant clones in the network analysis and also replicating the *IGHV3-23* gene. The first antibodies to be produced in a humoral immune response are always IgM and quickly progress to the production of all the different isotypes, IgD, IgA, IgG, and IgE but especial interest requires the IgD isotype[48]. IgD is co-expressed with IgM and secreted IgD exists and has a function in blood, mucosal secretions and on the surface of innate immune effector cells such as basophils[49]. Basophils are white cells that fight viruses, bacteria, parasites and fungi and they have been shown to be involved in renal diseases and transplant rejection[50]. Thus, another evidence that relates the B-cell responses to the rejection process with viruses and bacteria. Future studies are needed to demonstrate that this activation is due to differences in the recipient microbiome with implications in diagnostics and therapeutics.

This study allowed us to identify the relevance of some of these BCR clones, identify the BCR clones of clinical relevance with allograft rejection, and show that there is a likely relevance of these clones with heterologous immunity, given the enrichment of these clones in patients with colonized urinary tracts before transplant and their biological relevance to pathogen responses. A better understanding of the immune repertoire behavior in kidney transplant would not have been possible without the application of BCRSeq in combination with a robust implementation of a computational and statistical pipelines. Nevertheless, there are several limitations of our approach that should be recognized: First, our sample size is highly selected and relatively small in a set of pediatric patients and even though we reached statistical significance in our analysis and validated in two different sources of data, further analysis will be needed to corroborate our results. Secondly, our results are not exempted of the influence that biological and technical sampling may have in our analysis. Diversity measures are dependent of several issues: the fact that only a fraction of billions of cells in a repertoire are represented in a sample, differences in sequencing depth and possible experimental errors. We have extensively controlled for all these issues, first in the laboratory, running the same amount of blood for each sample in the same batch and second, computationally and statistically, applying recon tool to estimate the overall repertoire and performing downsampling to control for sequencing depth and experimental errors. Thirdly, we have performed BCRSeq using two different sources, gDNA and cDNA. Sequencing gDNA facilitates estimation of the clonality of a given Ig sequence since the number of sequence reads will be proportional to the number of gDNA molecules. Sequencing cDNA provides an estimate of the relative expression level of various Ig sequences in the repertoire. It has been shown that for T-cell receptors, the clonotype characterization performed on cDNA is not that good as in gDNA showing that the relative proportion of individual clones differed greatly[15] or that the tracking of HIV-specific clones is only successful with gDNA, but not mRNA[51]. Fourthly, the influence of immunosuppression may be a confounding factor on this type of analysis. In this study all the samples come from a clinical trial where the same immunosuppression load was received by all patients after transplantation. Nevertheless, we controlled by the two types of steroid-based and steroid-free to make sure that this was not an issue, observing no differences in the results. Finally, the cohort we present here is that of pediatric transplants. The majority of the transplanted clinical aspects are similar in children and adults. The immunosuppression and regimens used are similar, creatinine is the major serum biomarker, acute rejection is determined primarily by means of biopsy with the use of the Banff criteria and the rejection mechanisms of the kidney graft are generally similar[52], therefore the majority of the research done in adults or children may apply to both. However, other aspects such as non-compliance/non-adherence to immunosuppression, immunological aspects, the primary kidney diseases, leading to kidney failure, often associated with urologic issues, and the immunizations that are required before transplantation may differ, therefore further analysis in an adult population will be necessary to generalize these findings.

Despite these limitations, our data reveal that higher pre-transplant diversity is observed in individuals who go on to reject the organ, suggesting a predisposition of rejection, which may have future implications in predicting risk of rejection before engraftment, immunosuppression choices and clinical care practices. After 24 months of follow-up, an overall reduction in diversity over time accompanied by the persistence and expansion of certain clones and a higher use of several *IGHV* genes is observed in the rejection group, which may implicate particular common antigens in driving rejection. Special interest is observed for the increased use of *IGHV3-23* gene among the rejection patients since it has been previously associated with kidney transplantation and could be a key component to drive the rejection process. This work presents a longitudinal analysis of the B-cell immune repertoire in organ transplant promoting more studies to confirm these findings since they may have clinical implications in predicting, controlling, monitoring and treating kidney rejection.

## Methods

**Study design.** We studied 81 samples for gDNA and 56 matched samples for cDNA longitudinally at time 0, 6, and 24 months from a total number of 27 pediatric recipients who received a primary kidney transplant. The subjects in this study come from a clinical trial (SNSO1 multicenter study) where subjects were randomized (1:1) to a traditional low-dose steroid-based immunosuppression regimen (steroids, standard daclizumab induction until the second month post transplant, and maintenance immunosuppression with tacrolimus (Prograf, Astellas Pharma) and MMF (CellCept, Hoffman-La Roche) or a steroid-free immunosuppression regimen (prolonged daclizumab induction until the sixth month post transplant, tacrolimus and MMF). All subjects were enrolled following IRB approval and had informed consent. In this study, 14 patients received a steroid-avoidance regimen, while 13 received a steroid-based immunosuppressive regimen[53] and none of these patients received immunosuppression before transplant as this was one exclusion criteria. Treatment for rejection consisted of three pulses of intravenous corticosteroid (10 mg/kg) and baseline immunosuppression intensification.

All the samples used in the study had an associated serial allograft biopsy, which was read by a central pathologist, using semi-quantitative histological scores. Clinical acute rejection was defined as an acute rejection episode, associated with graft dysfunction, based on a greater than 10% rise in serum creatinine from baseline values, and confirmed through central pathological reading of the biopsies according to the updated Banff classification[23,24]. Chronic allograft injury was defined using the chronic allograft damage index (CADI) score. The patients were classified in three clinical outcomes defined by CADI score and rejection episodes Non-progressors (NP) had low non-incremental CADI score on three serial biopsies over 2 years, without acute rejection, progressors with no rejection (PNR) had higher CADI score on their serial biopsies over 2 years and incremental across the time points without rejection, and progressors with rejection (PR) had incremental high CADI scores on their serial biopsies over 2 years with rejection episodes. These patients were very carefully selected from a larger cohort of 120 patients, matching for demographic variables, and for all NP, and PNR to have no evidence inflammation on biopsy, or subclinical injury as measured by absent donor-specific antibodies. None had HLA or haplo- identical graft as this was an exclusion criteria for enrollment[54]. All samples were collected from 12 different US pediatric transplant programs between 2004 and 2006, under IRB approved protocols. The study was also approved by The Human Research Protection Program (HRPP) of the University of California, San Francisco and Stanford University to allow analysis of biobanked samples. All patients/guardians provided informed consent to participate in the research, in full adherence to the Declaration of Helsinki. The clinical and research activities being reported are consistent with the Principles of the Declaration of Istanbul as outlined in the Declaration of Istanbul on Organ Trafficking and Transplant Tourism.

**Isolation of gDNA and RNA**. Blood samples (4.5 ml) were collected into a 5 ml red top tube and incubated at room temperature for 30 min until the clot was formed. The sample was then centrifuged at $2000 \times g$ for 5 min using a swinging bucket rotor. The upper layer of serum was then transferred to another cryotube and the clot was stored in the same tube at $-80\,°C$ until use. Genomic DNA from whole-blood clot was extracted by using Clotspin Baskets and the Gentra PuregeneBlood Kit (Qiagen, Valencia, CA).

For RNA extraction from kidney needle biopsy (Qiagen, Valencia, CA) and stored at $-80\,°C$; total RNA was extracted using a master mix of 790 µl TRIzol and 10 µl glycogen. Tissue samples were homogenized, incubated at 15 to 25 °C for 5 min and 160 µl chloroform was added for phase separation. The mixture was incubated again at 25 °C for 2 min followed by centrifugation at 4 °C and used for RNA extraction using the RNeasy Micro Kit (Qiagen Catalog no.4004). RNA quantity and integrity were determined with the Thermo Scientific NanoDrop ND-2000 UV–Vis Spectrophotometer and Agilent Bioanalyzer, respectively.

**B-cell sequencing**. Genomic DNA templated PCR reactions were prepared from 100 ng gDNA aliquots to generate six independent barcoded libraries per sample. Multiplexed primers to the IgH J or FR1 or FR2 framework regions per the BIOMED-2 design were used[55]. Ten-nucleotide 'barcode sequences' in the primers were used to indicate the sample identity and replicate library identity for each PCR reaction. PCR was performed with AmpliTaq Gold (Roche) polymerase with the following program: 94 °C for 5 min; 35 cycles of (94 °C for 30 s, 60 °C for 45 s, 72 °C for 90 s); and final extension at 72 °C for 10 min. A second PCR reaction was carried out to ensure that libraries were not amplified to saturation prior to gel purification and sequencing. In all, 0.4 µl of each first PCR product templated the second PCR reaction using external primers specific for the 454 linker sequences; the amplification was carried out with the program: 94 °C for 15 min, 12 cycles of (94 °C for 30 s, 60 °C for 45 s, 72 °C for 90 s), and final extension at 72 °C for 10 min. The error rate of the AmpliTaq Gold should have a minimal effect on the identification of clonally related sequences or estimation of somatic mutation rates in sequences as discussed somewhere else[56]. cDNA was synthesized from total 300 ng of RNA with priming by random hexamers. Templates were amplified by PCR using Biomed IGHV primers in framework 1 (FR1) and isotype-specific primers located in the first exon of the constant regions. These primers[18,57] also encoded approximately half of the Illumina linker sequences needed for cluster generation and sequencing on the MiSeq instrument. Sample identity was encoded by eight-nucleotide multiplex identifier barcodes in each primer. For Illumina cluster recognition, four randomized nucleotides were encoded in the primers immediately after the Illumina linker sequence in the constant region primers. Each antibody isotype for each sample was amplified in a separate PCR reaction, to prevent formation of cross isotype chimeric PCR products. PCR was carried out with AmpliTaq Gold (Roche) following the manufacturer's instructions, and used a program of: 94 °C for 7 min, 35 cycles of (94 °C for 30 s, 58 °C for 45 s, 72 °C for 120 s), and final extension at 72 °C for 10 min. A second PCR step was used to add the remaining portion of the Illumina linkers to the amplicons and was carried out with the Qiagen Multiplex PCR kit (Qiagen) according to the manufacturer's instructions, using 0.4 microliters of the first PCR product as template in a 30 microliter reaction. The PCR program for the second PCR step was 94 °C for 15 min, 12 cycles of (94 °C for 30 s, 60 °C for 45 s, 72 °C for 90 s), and final extension at 72 °C for 10 min. The products of each PCR reaction were pooled in estimated equimolar amounts, electrophoresed on agarose gels, and gel extracted with QIAquick kits (Qiagen). High-throughput sequencing of genomic DNA templated libraries was performed on the 454 (Roche) platform using Titanium chemistry. cDNA library sequencing was performed on an Illumina MiSeq instrument using 600-cycle sequencing kits. A full list with the primers used with MiSeq (M154 and M155) and 454 Titanium (T7) are included in Supplementary Data 1.

Sequencing reads were processed as follows: paired-end reads were merged using FLASH[58] where after sequences were demultiplexed and trimmed of barcodes and IGHV primer sequences. The V, D, and J regions and V–D (N1), D–J (N2) junctions were identified using the alignment program IgBLAST[59]. Sequences were filtered to remove non-IGH artifacts, sequences with V gene insertion or deletions, chimeric sequences and nonfunctional sequences. At sample level, we excluded those with <100 clones (defined by same V and J segments, same CDR3 length and 90% nucleotide identity) as a control for bad quality samples. After quality control, for gDNA we had complete longitudinally data for 69 samples at time 0, 6 and 24 with a total number of 327,703 reads (mean 4045 per sample). For cDNA, we had complete data for 55 matched samples, although no time 0 samples were further available. In this case, we had isotype-specific information (1,773,330 reads for IgD (31,667 per sample), 1,708,227 reads for IgM (30,504 per sample), 973,444 reads for IgA (17,383 per sample), 139,7345 reads for IgG (24,953 per sample), and 29,000 reads for IgE (5,178 per sample).

**Diversity analysis**. Diversity is measured by species richness considering the number of clones per sample. This measure does not consider the frequency of each species, so we also used Shannon entropy ($H$) to measure diversity providing information about the size distribution of species in the population. $H$ is defined as:

$$H = -\sum_{i=1}^{N} p_i log_2 p_i \tag{1}$$

where $N$ is the number of unique clones and $p_i$ is the frequency of clone $i$. $H$ ranges from 0 (sample with only one clone) to $H\max = \log_2 N$ (sample with a uniform distribution of clones).

Then, we used general linear model to find the association between richness and entropy with the clinical outcome at the different time points. We adjusted this model by all the clinical variables available showed in Table 1 to be sure that any of the characteristics of the patient was a confounding factor.

To model the longitudinal component of the data, we applied linear-mixed effect model considering a conditional growth model as shown in Supplementary Fig. 10. To apply this model, we used lme4 package in R considering the interaction between clinical outcome and time to find association with richness and diversity with time being a random effect.

To deal with the fact that diversity measures may be affected by the missing species problem (only a fraction of billions of cells in a repertoire are represented) and sequencing and experimental errors, we performed two strategies in addition of the full data analysis. First, we used Recon (reconstruction of estimated clones from observed numbers) tool[29] to deal with the missing species problem. Recon is a modified maximum likelihood method that outputs the overall diversity of a repertoire from measurements on a sample. Recon outputs accurate and robust estimates of a set of diversity measures, including richness and entropy allowing robust comparisons of diversity between individuals. Second, we performed a downsampling strategy taking a random subset of reads for each sample equal to the smallest sequencing size, followed by the re-calculation of the B-cell clones to adjust by sequencing depth and deal with possible experimental errors. For gDNA, there are samples with very low reads ($< 1000$), and to avoid losing many sequences and the reality of the data, we excluded a total of nine samples that had <1000 reads. In the case of cDNA, this was not necessary. We generated ten random subsamples to account for possible stochastic effects and performed the diversity analysis at each time point and the longitudinal data analysis on the mean value of ten independently downsampled diversity estimates.

**Network analysis**. The network generation algorithm is very similar to the one defined before[31]. Briefly, each vertex represents a B-cell sequence where the size is defined by all the identical sequences. Edges are calculated using the clone definition (same V and J segments, same CDR3 length and 90% nucleotide identity between CDR3s) and clusters represents each clone in the repertoire. The analysis was done using igraph package in R using the layout_with_graphopt option to generate the plot.

To quantify the network, we calculated the Gini Index for vertex size and cluster size. Gini Index is a measure of unevenness extensively used to measure wealth distribution. It measures the inequality among values of frequency distribution. We used the Gini function from ineq package in R to calculate the Gini coefficient for vertex size and cluster size distribution. A Gini coefficient of zero expresses perfect equality and a Gini coefficient of 1 expressed maximal inequality.

**Clonal analysis**. To study the specific clones associated with clinical outcome, we built a matrix with all clones that are present in more than one sample (total number = 118,223) by all samples at each time point. We then studied the association with clinical outcome of each particular clone defining one variable per clone as present/no present. Finally, we applied Fisher's exact test to account for significance in the $2 \times 3$ table for each clone in each time point. We also studied the persistence clones defined by those clones that are in more than one time point in each individual. Then, to account for differences by clinical outcome, we applied a linear model defining as dependent variable the number of clones (to measure differences in persistence) and the number of counts of each clone (to measure the clonal expansion) and clinical outcome as independent predictor.

***IGHV* gene usage analysis**. To study the *IGHV* genes, we assessed the *IGHV* gene usage per sample as the number of times each gene is used normalized by the number of clones to avoid overrepresentation of certain *IGHV* genes. For statistical analysis, we filtered out those genes with very low expression (IGHV usage/clones $< 0.05$) in at least 10% of the samples. In total, we analyzed 27 *IGHV* genes corresponding to 63 samples. Then, we applied a linear model to find those genes that were associated with clinical outcome in each time point and corrected these results by multiple testing using Benjamini and Hochberg FDR < 0.05. We adjusted this model by all the clinical variables available showed in Table 1 to be sure that any of the characteristics of the patient was a confounding factor.

**Reporting summary**. Further information on research design is available in the Nature Research Reporting Summary linked to this article.

## Data availability

The data supporting this publication has been deposited in the ImmPort repository under the study accession SDY1361(https://www.immport.org/shared/study/SDY1361). All other data are available from the authors upon request. Source data for the graphs and statistical analyses can be found in the Source Data file.

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

## Acknowledgements

We thank Scott Boyd and Krishna Roskin from the Boyd's lab for generating the data and contributing in the discussion of the data. We thank Mikel San Vicente, senior software engineer, for his contribution with the coding to define the clones and perform the network visualization. We acknowledge funding support from U01AI113362-01 (M.M.S) and P30AR070155 (M.S.). M.S. is supported by K01P0511836.

## Author contributions

M.M.S., M.S. and S.P. conceived the study design and analysis plan. M.S., M.M.S. and S.P. carried out the bioinformatics/data analysis plan. S.P. performed the data analysis and generated all figures and tables. J.M.L., F.V., T.K.S and M.M.S. collected and analyzed clinical materials. M.S. and M.M.S. supervised the work.

## Additional information

**Competing interests:** The authors declare no competing interests.

