## [Peer Review File · Nature Communications]

Reviewers' comments:

Reviewer #1 (Graft rejection, allo-response)(Remarks to the Author):

This paper investigates associations between B cell repertoire as determined by high throughput IgH CDR3 sequencing and outcomes in a group of 27 pediatric kidney transplant recipients who are studied longitudinally. Significant associations are seen between high pre-transplant BCR repertoire diversity and rejection by 2 years and between loss of diversity post-transplant and rejection. The studies are of interest and of considerable novelty, but further analyses connecting the shared sequences to cause of nephropathy and outcomes could enhance the paper.

Major concerns:

Shared sequences and increased usage of IGHV3-23 are detected among rejecting patients. However, no indication is given of the V gene usage of the shared sequences- is IGHV3-23 overrepresented?

A connection between IGHV3-23 and responses to bacterial superantigens is discussed, but no indication is given as to whether or not expansions of this family prior to transplant were associated with potentially infectious causes of renal failure in the population studied and whether or not bacterial infection was a possible trigger of rejection episodes. Additional information could help to cement this connection. Furthermore, an indication of the cumulative frequencies of the shared sequences before and after transplantation would be useful for evaluation of their significance.

Overall, the basis for the premise that shared clones would be detected in patients with rejection is unclear. Shared clones could imply a common alloantigen. Since the HLA disparities were different among different patients, one would not anticipate that DSAs against the same antigen would be made. Was there an association between these shared clones and the development of specific DSAs? Data should be provided on DSA development in relation to all of the results. If the authors have reason to think that DSAs are not the relevant targets of the shared clones, this should be mentioned and could perhaps be solidified by further information regarding infection as discussed above.

Additional comments:

- 1) The paper could benefit from greater emphasis on the fact that the patients were all pediatric, as their causes of ESRD may be different from those in adults and the results may not be generalizable to adults;
- 2) The abstract and other sections of the paper imply that the expanded B cells in patients with rejection are alloimmune, but no evidence is provided to indicate that this is the case. The abstract and other sections should be modified accordingly;
- 3) The basis for the statement in the Abstract that "peripheral activation of the B-cell immune repertoire was noted" is unclear. Overexpression of several IGHV genes cannot be described as evidence of B cell activation;
- 4) The authors do not propose an explanation for the association of greater pre-transplant BCR diversity with increased rejection;
- 5) The statement on lines 195-196 that both recon and down-sampling replicated the associations of rejection with diversity is an oversimplification. For example, the significance of pre-transplant diversity associating with rejection is lost with recon (fig S4B) and downsampling (S4F and H). Other associations are lost as well. These results are brushed over in the text;
- 6) It is counterintuitive that patients with greater repertoire diversity prior to transplant had more clonal expansions. Clonal expansions should reduce repertoire diversity. Could the diversity measurements be failing to capture this? Some discussion of this point is needed;
- 7) Line 268 refers to "higher number of persistent clones" with a p value of 0.09. This statement should not be made on the basis of statistically insignificant data;

8) The statement in lines 274-277 is puzzling. Of course there would be differences in repertoires between individuals. What is the meaning of the statement that there would be “subtle, and currently unmeasured, differences in B cell repertoire”?

9) Lines 393 and 435- the reference to “huge implications” of these findings seems premature. The implications seem unclear at this stage.

Minor comments:

1) Line 54 refers to sequencing of “millions of Abs”. This should be changed to reflect that Ig regions in cells are sequenced, not antibodies themselves.

2) Line 119 and further down the page- it should be indicated that the numbers in brackets “(mean 4,045/sample)” refer to numbers of unique sequences;

3) I am puzzled by the repeated reference to the patient with PTLD, who demonstrated increasing B cell clonal diversity after KTx. Wouldn't PTLD be an oligoclonal or monoclonal expansion that would decrease repertoire diversity?

Reviewer #2 (Kidney transplant)(Remarks to the Author):

This is a novel study in transplant recipients examining the B cell repertoire using high throughput sequencing (BSeq). Previous attempts to examine this issue in the 1990's have employed much cruder methods, but have also demonstrated potential clinically relevant signals. It is thus of considerable interest to the field, for the reasons stated in the introduction to this communication, to determine whether or not BSeq can be predictive of outcome in renal transplant recipients.

I am not a biostatistician and warn that I am not qualified to comment extensively, or in any detail, on the bio-statistics used in this study.

The background for the study of B cell biology in renal transplant recipients has been most well characterised in small numbers of long term tolerant renal recipients. There have been 20 – 30 individuals identified who had discontinued immunosuppression post transplant and not rejected their grafts after prolonged follow up periods. These patients are spread across N America and Europe and have been studied extensively. The predominant message from these studies is that the tolerant patients have different B Cell repertoire to non-tolerant patients. The problem has been with interpretation of these data that the tolerant patients were not on drugs that influence immunity while the non-tolerant were. This remains the critical weakness in these studies – are we observing the effect of pharmacology or important biological diversity? This study may have some of the same issues.

The predominantly clinical issues which are not clarified in this study are:

1. The cohort is highly selected and quite small in number. They are paediatric patients from the period 2000-2011 (ref 25) with biobanked samples.

a. The cause of renal failure is categorised, but it is not clear if any of the patients were given steroids or other immunosuppressants pre-transplant. If even two or three were, then this could skew the important and tantalising pre-transplant results demonstrated in this study.

b. Immunosuppression post-transplant is not detailed but includes Tacrolimus, Mycophenolate Mofetil and variable steroids. No treatment for rejection is detailed, yet Progressors had protocol biopsies showing inflammation which was potentially if not likely treated at the time. Patients with fibrosis but

no inflammation (NPR) would more likely have had tacrolimus reduced and stable patients would have had no alteration or perhaps general reduction in immunosuppression. These alterations may be of relevance to the variables studied. The repeated statement that no variations were found based on age, gender, immunosuppression, etc gives this reader little confidence, since the n=7 in the critical group is too small to even test those variables legitimately. How many patients received, either at induction or for treatment of rejection depleting monoclonal or polyclonal antibodies? Certainly some received ATG which is not as specific to T cells as the product name suggests and thus may have influenced the BSeq results.

c. The patients received mixed Living and Deceased donor organs, with variable levels of HLA mismatch and reported absence of Donor Specific Antibodies (DSA). The problem here is that unless the donors and recipients were retyped and serum retested recently for this study, the data is relying in part at least, on old and quite imprecise technology from the early 2000's. There were no HLA identical grafts reported. Some of the Living donors may have had haplotype matches, which might distort the B cell response, but this is not reported.

d. The B cell response of 1 year olds and 15 year olds may well be different, but there are insufficient numbers to examine that question.

e. The word 'rejection' is used rather loosely through the paper. It is highly relevant to be terminologically specific. Rejection episodes occurring early post transplant and treated successfully are not detailed. Graft losses were presumably all beyond 2 years since all samples had protocol histology available. In the discussion the reader is informed that rejections occurred after point 6 (6 months) – what is the definition of that word in this paragraph? When did clinical rejection occur? When did graft failure occur? What do the authors mean by rejection?

2. The most frequent cause of rejection (immunological alloimmune damage to the graft) in the long term renal allograft in teenagers is medication non-compliance/non-adherence. Thus the likely association of the status 'PR' and under immunosuppression. Is BSeq thus potentially simply a very complicated way of seeing if patients are taking their immunosuppressant medications? I don't really believe this to be the case but the issue does need discussion.

3. Since the transplants were apparently performed up to nearly two decades ago, the long term outcomes are certainly known. Why were these not detailed?

4. A single 'PR' individual appears to have a 30 month time point plotted in most of the data figures. Why was this single point plotted and what difference would it make to the statistical analyses to exclude this individual? It certainly seems that this one individual might skew the results in most of the plots.

In summary, application of the technique of BSeq to the renal transplant population may have potential to determine relevant differences, even pre-transplant, that may influence outcomes and which, importantly, may be amenable to intervention. This study of a small mixed population of paediatric patients demonstrated that pre-transplant differences existed in groups of patients with different outcomes at the two year time point. It is, however, hard to be convinced, on these data, that the post transplant differences related to histological outcomes are as clear as proposed.

The authors have made a good case for the wider and more extensive use of these techniques - both biological and biostatistical - and hold out the hope that there are relevant and important differences in individuals which explain and predict the heterogeneous response to kidney transplantation.

There is a number of minor issues with grammar and proofing of the paper, such as random use of 'Abs' and a section of type in superscript, all of which are easily fixed.

Reviewer #3 (BCR repertoire, epigenetic)(Remarks to the Author):

NCOMMS-18-18903

Mismatching at non-HLA loci, may influence the immune response of the recipient against the kidney donor graft and result in poor long-term graft outcomes. In this work, the authors have used high throughput BCR sequencing approach to analyze the BCR diversity and clonality in 27 kidney transplantation recipients, including non-progressors (NP; n=10), progressors with no rejection (PNR; n=10) and progressors with rejection (PR; n=7), before and 6 or 24 months after kidney engraftment. The authors conclude that individuals who developed rejection after kidney transplantation had a more diverse Ig VHDJH repertoire before the transplant than those who do not. Although it may be important to understand the impact of B-cell diversity on kidney transplant rejection, the putative differences in Ig VHDJH repertoire between NP and PNR or PR showed in the manuscript are likely resulted from experimental limitation.

The major conclusion of this manuscript is that individuals who developed rejection after kidney transplantation had a more diverse repertoire before the transplant. This is largely based on the results in Fig. 2 showing that the numbers of Ig VHDJH clones (with unique sequences) from NP samples were smaller than those from PNR or PR samples. However, as noticed by the authors, diversity measures may be affected by biological and technical sampling, and only a fraction of billions of B cells are represented. As indicated in Fig. S1, the lower numbers of clones from the NP samples were correlated to a much fewer sequence reads generated from NP genomic DNA, as compared to PNR and, more significantly, PR. The numbers of sequence reads can be largely affected by several technical issues, such as PCR amplification to generate amplicons for miSeq sequencing, the amount and quality of genomic DNA, etc. A more accurate measurement would be the percentage of clones among the total reads from each individual sample. Indeed, as showed in Fig. 4 and Fig. S1, the proportion of clones among the total reads from PNR and PR samples are much lower than that from NP samples. At time 0, the VHDJH clones account of $789/1062=74.3\%$ of total reads from NP individual 1, while the VHDJH clones account of only $1581/4171=37.9\%$ and $4496/15495=29.0\%$ of total reads from PNR individual 20 and PR individual 25, respectively. This conclusion is the opposite to that made by the authors.

In addition, the conclusion that "after 2 years of follow-up, patients who developed rejection showed an overall reduction of peripheral B-cell diversity" likely resulted from the reduced total reads from 24 PR samples (may be partially due to one less sample at this time point as compared to time 0, as indicated by Fig. S1) and increased total reads from time 24 NP samples. The increased total reads could be simply due to better PCR amplifications. Further the increase in diversity in NP across time after kidney transplantation, as emphasized in page 9, was likely also due to higher total reads from samples of time 6 and time 24 as compared to that of time 0.

Addition points:

1. Allograft rejection is associated with development of functional IgE, especially IgE specific for donor MHC antigens. It would make more sense to understand the diversification of IgE BCR.
2. It would be important to know SHM levels in the three groups.
3. The amplicons used for sequences were amplified with AmpliTaq Gold, which is not a high fidelity polymerase and generates a high error-rate.
4. The sequence of the PCR primers should be included.
5. A Principal Component Analysis to determine whether the repertoires in each group are similar while different from other groups should be included.

Reviewer #4 (Biostat tech referee):

I'm confused about the diversity analysis. It seems there are 3 diversity analyses, 1) the full data, 2) the recon based diversity analysis and 3) the down-sampled strategy. Can you provide more details on the dis/agreement between the results from these methods?

How was the FDR adjustment implemented in the IGHV analysis? E.g., was it Benjamini and Hochberg?

We appreciate the valuable comments from the editors and reviewers and believe we have addressed all of the reviewers' concerns in the new manuscript with further explanations and analysis. We have tracked the changes in the main text. Please see our point-to-point responses in bold below.

Reviewers' comments:

Reviewer #1 (Graft rejection, allo-response)(Remarks to the Author):

This paper investigates associations between B cell repertoire as determined by high throughput IgH CDR3 sequencing and outcomes in a group of 27 pediatric kidney transplant recipients who are studied longitudinally. Significant associations are seen between high pre-transplant BCR repertoire diversity and rejection by 2 years and between loss of diversity post-transplant and rejection. The studies are of interest and of considerable novelty, but further analyses connecting the shared sequences to cause of nephropathy and outcomes could enhance the paper.

We would like to thank the reviewer for finding our work novel and interesting. We agree that connecting the shared sequences to the cause of graft failure would be of interest. When we evaluate different causes of graft failure we do not find any specific association with pre-transplant differences in BCR repertoire diversity. Previous studies by our group, evaluating this similar question in the context of immunogenicity, also did not demonstrate any impact of cause of kidney failure and variations in pre-transplant immune responses (Butte et al, MCP, 2011; PMID: 21183621)

Major concerns:

1) Shared sequences and increased usage of IGHV3-23 are detected among rejecting patients. However, no indication is given of the V gene usage of the shared sequences- is IGHV3-23 overrepresented?

This is a good point to consider. We have performed an enrichment analysis using Fisher's exact test showing that the IGHV3-23 sequences were significantly over-represented in both, the clones shared among individuals in the persistence clones (p-value $<2.2 \times 10^{-16}$), and the clones associated with clinical outcome at time 24 (p-value $<2.2 \times 10^{-16}$). We have added this information in lines 311-316.

2) A connection between IGHV3-23 and responses to bacterial superantigens is discussed, but no indication is given as to whether or not expansions of this family prior to transplant were associated with potentially infectious causes of renal failure in the population studied and whether or not bacterial infection was a possible trigger of rejection episodes. Additional information could help to cement this connection.

The cause of renal failure was considered in all the analysis, including the IGHV3-23 analysis. We have included a horizontal bar in the heatmaps from figure 6 showing each individual ESRD and there is no connection with rejection. Nevertheless, since the numbers were very small to perform any statistical analysis with this variable, we grouped the ESRD into three main groups for further evaluation (Non-immune/Structural, Reflux and Other/Unknown). This variable was not associated with IGHV3-23 expression, although there was a borderline p-value for reflux disease. We adjusted the IGHV3-23 expression analysis by this variable and the results remain

significant, although we observed that the NP misclassified patient in the analysis (Fig. 6) had a reflux disease. This is discussed in the article (lines 396-399)

3) Furthermore, an indication of the cumulative frequencies of the shared sequences before and after transplantation would be useful for evaluation of their significance.

The number of persistent clones shared among patients is limited. There are only 23 clones shared across time with a total of 1051 sequences. Even though we believe this indicator may not be the most reliable due to the small number of shared clones, the cumulative frequency is obtained for the three clinical outcomes and we observe that the PR has a higher incidence of shared sequences than the others post-transplant, and it is also higher pre-transplant:

	NP	PNR	PR
pre	0.05	0.10	0.18
post	0.23	0.31	0.46

4) Overall, the basis for the premise that shared clones would be detected in patients with rejection is unclear. Shared clones could imply a common alloantigen. Since the HLA disparities were different among different patients, one would not anticipate that DSAs against the same antigen would be made. Was there an association between these shared clones and the development of specific DSAs? Data should be provided on DSA development in relation to all of the results. If the authors have reason to think that DSAs are not the relevant targets of the shared clones, this should be mentioned and could perhaps be solidified by further information regarding infection as discussed above.

Only one individual in the PNR group (individual 16) developed DSA class I and another individual in the PR group (individual 25) had also class I DSA at time 0. The others did not have at time 0 or develop later on time any DSA. We have included this information in table 1 and have adjusted the analysis for this specific variable to make sure that this was not influencing the results. We obtain similar p-values. This additional and new analysis is now included in the revised paper.

Additional comments:

1) The paper could benefit from greater emphasis on the fact that the patients were all pediatric, as their causes of ESRD may be different from those in adults and the results may not be generalizable to adults;

In the last point of the limitations in the discussion (lines 460-464), we address this issue highlighting the differences and similarities between a pediatric cohort and adult cohort. We have clarified the paragraph proposing that further analysis need to be done to generalize these findings to an adult population.

2) The abstract and other sections of the paper imply that the expanded B cells in patients with rejection are alloimmune, but no evidence is provided to indicate that this is the case. The abstract and other sections should be modified accordingly;

The assumption is that the expanded clones are alloimmune as the identified B cell expansion is correlated with the phenotype of development acute rejection by biopsy confirmed, which is an alloimmune event. Nevertheless, we have altered the language in the abstract and other sections of the paper and added to the discussion (lines 379-382) that even though we can hypothesize that the expansion of these clones is likely linked to alloimmune injury to the graft, direct evidence that the expansion of these clones are alloimmune requires additional in vitro and animal studies.

3) The basis for the statement in the Abstract that “peripheral activation of the B-cell immune repertoire was noted” is unclear. Overexpression of several IGHV genes cannot be described as evidence of B cell activation;

We apologize for the confusing wording here. We have modified the statement in the abstract to just speculate that the IGHV gene overexpression may be an indicative of B-cell activation.

4) The authors do not propose an explanation for the association of greater pre-transplant BCR diversity with increased rejection;

Cellular and humoral immunity pathways are indispensable for combating numerous infections and for controlling alloimmunity. Protective and responsive capacities of T and B cells involved in heterologous and innate immunity are defined not only by the amount and phenotype but also by the BCR and TCR repertoire diversity allowing targeting of different alloimmune and pathogenic epitopes. This study allowed us to identify the relevance of some of these BCR clones, identify the BCR clones of clinical relevance with allograft rejection, and show that there is a likely relevance of these clones with heterologous immunity, given the enrichment of these clones in patients with colonized urinary tracts before transplant and their biological relevance to pathogen responses. We have added a paragraph in the discussion to comment on this (lines 427-431)

5) The statement on lines 195-196 that both recon and down-sampling replicated the associations of rejection with diversity is an oversimplification. For example, the significance of pre-transplant diversity associating with rejection is lost with recon (fig S4B) and downsampling (S4F and H). Other associations are lost as well. These results are brushed over in the text;

We apologize if some of the comparisons were “brushed over in the text”. We have now added a more comprehensive description (lines 204-216) of which associations remain and which are lost. We thank you for enhancing the clarity of the text with this suggestion.

6) It is counterintuitive that patients with greater repertoire diversity prior to transplant had more clonal expansions. Clonal expansions should reduce repertoire diversity. Could the diversity measurements be failing to capture this? Some discussion of this point is needed;

Thank you for pointing out this potential discrepancy. Indeed, we observe a higher number of clones pre-transplant in patients who go on to reject the organ, which are those that shows a decrease in diversity along the time due to the clonal expansion as observed in the longitudinal analysis (Figure 3). So, we observed higher diversity as shown by several measures that we apply, and we also observe that several of those clones are highly

expanded, which doesn't necessarily contradict the increased diversity pre-transplant. We have clarified this point in the first paragraph of the discussion.

7) Line 268 refers to "higher number of persistent clones" with a p value of 0.09. This statement should not be made on the basis of statistically insignificant data;

Thank you for pointing this out. We have modified the statement with the addition of: "showing a trend, although non-significant, with a higher number of persistent clones (p-value = 0.09)"

8) The statement in lines 274-277 is puzzling. Of course there would be differences in repertoires between individuals. What is the meaning of the statement that there would be "subtle, and currently unmeasured, differences in B cell repertoire"?

Thanks for this comment. We agree and have deleted this sentence since we do not think it adds to the message.

9) Lines 393 and 435- the reference to "huge implications" of these findings seems premature. The implications seem unclear at this stage.

Thank you for the suggestion. We have modified the reference to "important implications" and rephrase the paragraph (Line 423-426).

Minor comments:

1) Line 54 refers to sequencing of "millions of Abs". This should be changed to reflect that Ig regions in cells are sequenced, not antibodies themselves.

Thank you, this has been corrected.

2) Line 119 and further down the page- it should be indicated that the numbers in brackets "(mean 4,045/sample)" refer to numbers of unique sequences;

We apologize for the confusion - these are actually not unique sequences, but the number of reads; we could have the same sequence in two reads.

3) I am puzzled by the repeated reference to the patient with PTLN, who demonstrated increasing B cell clonal diversity after KTx. Wouldn't PTLN be an oligoclonal or monoclonal expansion that would decrease repertoire diversity?

Sorry for the confusion. We do observe an increase in B cell clones at time 6 and a decrease on clones and diversity at time 24 showing that there is a clonal expansion of specific clones that decrease the repertoire diversity along the time. We have clarified this in the paper.

Reviewer #2 (Kidney transplant)(Remarks to the Author):

This is a novel study in transplant recipients examining the B cell repertoire using high throughput sequencing (BSeq). Previous attempts to examine this issue in the 1990's have employed much cruder methods, but have also demonstrated potential clinically relevant signals. It is thus of considerable interest to the field, for the reasons stated in the introduction

to this communication, to determine whether or not BSeq can be predictive of outcome in renal transplant recipients.

We would like to thank the reviewer for recognizing the value of our study.

I am not a biostatistician and warn that I am not qualified to comment extensively, or in any detail, on the bio-statistics used in this study.

The background for the study of B cell biology in renal transplant recipients has been most well characterised in small numbers of long term tolerant renal recipients. There have been 20 – 30 individuals identified who had discontinued immunosuppression post transplant and not rejected their grafts after prolonged follow up periods. These patients are spread across N America and Europe and have been studied extensively. The predominant message from these studies is that the tolerant patients have different B Cell repertoire to non-tolerant patients. The problem has been with interpretation of these data that the tolerant patients were not on drugs that influence immunity while the non-tolerant were. This remains the critical weakness in these studies – are we observing the effect of pharmacology or important biological diversity? This study may have some of the same issues.

Thank you for the background of the field as well as bringing up the issue of pharmacology and the effects that we might be observing here. We agree that the effect of immunosuppression could be observed here, but the patients studied in this analysis come from a clinical trial and all of them received the same immunosuppression load after Ktx. In the discussion in line 368 we address this issue and we provide that a possible explanation of the decrease on diversity in the patients who develop rejection, may relate to the fact that these patients receive a temporarily greatly increased load of immunosuppression for the treatment of rejection. This could explain the reduction in B-cell diversity over time, and explain the difference at 24 months post-Ktx, but this is unlikely to be the cause, as the reduced diversity in the PR group is also seen at 6 months post-Ktx, at a time when many patients had not yet developed rejection. In addition to this, we have controlled all our studies by immunosuppression to control by the differences in steroid-based and steroid-free, showing non-confounding effect on this regard. Even though we comment this issue along the discussion, we have added a paragraph in the limitations (line 451-455), to make sure that this issue is addressed properly in the text.

The predominantly clinical issues which are not clarified in this study are:

1. The cohort is highly selected and quite small in number. They are pediatric patients from the period 2000-2011 (ref 25) with biobanked samples.

Reference 25 was incorrectly cited, and we apologize for this - this has now been corrected. Patients were pediatric samples (aged 0 to 21 years) selected between 2004-2006 as defined in (Sarwal et al, 2012 AJT, PMID:22694755). The same study design and protocol for treatment was used since this protocol (Li et al, 2009, AJT, PMID: 19459814) was the one tested in the clinical trial defined in Sarwal et al, 2012 AJT. We apologize for the wrong citation. This has been corrected in the text.

We also recognize that the patients here are highly selected and the number of patients is rather small, however this is meant to be a pilot study which will hopefully lead to larger

follow up experiments both in pediatric and adult populations. We have emphasized this point in the limitation section on lines 434-435.

a. The cause of renal failure is categorised, but it is not clear if any of the patients were given steroids or other immunosuppressants pre-transplant. If even two or three were, then this could skew the important and tantalising pre-transplant results demonstrated in this study.

That is a very important point to consider on this type of studies, but none of these patients received immunosuppression before transplant since this was one exclusion criteria of the original clinical trial. We have clarified this in materials and methods.

b. Immunosuppression post-transplant is not detailed but includes Tacrolimus, Mycophenolate Mofetil and variable steroids.

Subjects of this study come from a clinical trial where subjects were randomized (1:1) to a traditional low dose steroid-based immunosuppression regimen (steroids, standard daclizumab induction until the second month post-transplant, and maintenance immunosuppression with tacrolimus (Prograf, Astellas Pharma) and MMF (CellCept, Hoffman-La Roche) or a steroid-free immunosuppression regimen (prolonged daclizumab induction until the sixth month post-transplant, tacrolimus and MMF).

We have extended this information in materials and methods to better clarify the immunosuppression regimen and more detailed information can be read on the ref 25 (Sarwal et al, 2012 AJT)

No treatment for rejection is detailed

Sorry about this, treatment for acute rejection consisted of three pulses of intravenous corticosteroid (10 mg/kg) and baseline immunosuppression intensification. This has been added to material and methods.

Progressors had protocol biopsies showing inflammation which was potentially if not likely treated at the time.

They were not treated for this subclinical inflammation as treating this was not part of the study protocol.

Patients with fibrosis but no inflammation (NPR) would more likely have had tacrolimus reduced and stable patients would have had no alteration or perhaps general reduction in immunosuppression.

Though these suggested changes would make sense, these were not undertaken in this clinical trial as this was not the study protocol. Hence these treatment variables do not apply in this study and analysis.

These alterations may be of relevance to the variables studied. The repeated statement that no variations were found based on age, gender, immunosuppression, etc gives this reader little confidence, since the n=7 in the critical group is too small to even test those variables legitimately. How many patients received, either at induction or for treatment of rejection depleting monoclonal or polyclonal antibodies? Certainly, some received ATG which is not as specific to T cells as the product name suggests and thus may have influenced the BSeq results.

All patients in this analysis received steroid pulse intensification only for treatment of acute allograft rejection. This has now been included in material and methods.

c. The patients received mixed Living and Deceased donor organs, with variable levels of HLA mismatch and reported absence of Donor Specific Antibodies (DSA). The problem here is that unless the donors and recipients were retyped and serum retested recently for this study, the data is relying in part at least, on old and quite imprecise technology from the early 2000's.

All of these patients had detailed antibody testing performed at the Terasaki Lab later- Please see the reference to the paper with all of this data (Chaudhuri et al 2013, JASN PMID: 23449533).

There were no HLA identical grafts reported.

Yes, there were no HLA identical graft as this was an exclusion criterion for enrollment in this clinical trial. A sentence has been added to materials and methods.

Some of the Living donors may have had haplotype matches, which might distort the B cell response, but this is not reported.

There were no HLA or haplo-identical transplants in this study. This has now been added to materials and methods.

d. The B cell response of 1 year olds and 15 year olds may well be different, but there are insufficient numbers to examine that question.

This is an excellent point and we are very interested in exploring how the B cell repertoire and response changes with age. While as the reviewer mentions we don't have the numbers to study this question with the existing pilot cohort, this is something that we aim to explore in the future.

Nevertheless, to make sure that the age was not an issue here, in addition of adjusting by age all our analysis, we have performed a sensitivity analysis excluding two individuals the minimum (1-year-old) and the maximum (19 years old). The results remain significant (NP vs. PR at time 0: richness: p-value = 0.01, entropy: p-value = 0.03). We have added these analyses to the result section in line 158:161.

e. The word 'rejection' is used rather loosely through the paper. It is highly relevant to be terminologically specific.

Clinical acute rejection was defined as an acute rejection episode, associated with graft dysfunction, based on a greater than 10% rise in serum creatinine from baseline values, and confirmed through central pathological reading of the biopsies according to the updated Banff classification (ref 23 and 24). Further detail can be seen in ref 25. We have extended this information in materials and methods.

Rejection episodes occurring early post transplant and treated successfully are not detailed. All rejection occurred in the first year post-transplant; the clinical trial had 3 year followup.

All rejections in this subset has a rejection time of 12.5 (7.0) months (mean(sd)). All of the rejection episodes occurred after 6 months of follow-up and they were treated with steroid pulse intensification and had complete/partial recovery of function over the following months of follow-up. We have added this information to Table 1.

Graft losses were presumably all beyond 2 years since all samples had protocol histology available.

Yes, that is correct.

In the discussion the reader is informed that rejections occurred after point 6 (6 months) – what is the definition of that word in this paragraph? When did clinical rejection occur? When did graft failure occur? What do the authors mean by rejection?

Thank you for all these suggestions, they were not properly reflected in the paper. We have addressed them above and incorporated into the manuscript.

2. The most frequent cause of rejection (immunological alloimmune damage to the graft) in the long term renal allograft in teenagers is medication non-compliance/non-adherence. Thus the likely association of the status ‘PR’ and under immunosuppression. Is BSeq thus potentially simply a very complicated way of seeing if patients are taking their immunosuppressant medications? I don't really believe this to be the case but the issue does need discussion.

We have included in the discussion this possible limitation (461), although this wouldn't be the case for time 0 observation. Besides, there are no objective measures that these individuals were non-adherent based on regular clinic and study visits and labs and target drug trough levels.

3. Since the transplants were apparently performed up to nearly two decades ago, the long term outcomes are certainly known. Why were these not detailed?

These patients were enrolled from 12 different transplant programs and all patients have not been followed up at the different centers after the NIH study was completed at 3 yrs follow-up. Getting additional data on follow-up though desirable is not available.

4. A single ‘PR’ individual appears to have a 30 month time point plotted in most of the data figures. Why was this single point plotted and what difference would it make to the statistical analyses to exclude this individual? It certainly seems that this one individual might skew the results in most of the plots.

This is an important point. We have one individual that appears in the longitudinal analysis (the rest of the analysis are considering only those samples at 0, 6 or 24 months) with an extra sample at 32 months post-transplant. We agree completely that this individual might skew the results. For that reason, we have performed a sensitivity analysis excluding this particular sample. For gDNA and IgD isotypes, the results remain the same. Only IgG and IgM isotype were borderline significant. We have added this sensitivity analysis to the paper in line 182-186.

In summary, application of the technique of BSeq to the renal transplant population may have potential to determine relevant differences, even pre-transplant, that may influence outcomes

and which, importantly, may be amenable to intervention. This study of a small mixed population of paediatric patients demonstrated that pre-transplant differences existed in groups of patients with different outcomes at the two-year time point. It is, however, hard to be convinced, on these data, that the post-transplant differences related to histological outcomes are as clear as proposed.

Pre-transplant differences are one of the main conclusions of the paper as well as the fact that B-cell sequencing can be used to monitor what is happening through the process of transplantation. We agree that more studies need to be done to corroborate these findings and as suggested by the reviewers, we have toned down the paper and focus on the fact that this is a pilot study. We also highlight that our study provides a much larger cohort than the only other paper published on B cell sequencing for the study allograft rejection, as that study had only 12 participants (Vollmers et al, Plos Medicine, 2015, PMID: 26466143)/

The authors have made a good case for the wider and more extensive use of these techniques - both biological and biostatistical - and hold out the hope that there are relevant and important differences in individuals which explain and predict the heterogeneous response to kidney transplantation.

Thank you - this was one of the major goals of our work and we hope to continue with larger studies moving forward.

There is a number of minor issues with grammar and proofing of the paper, such as random use of 'Abs' and a section of type in superscript, all of which are easily fixed.

Thank you, we have proof read and tried to correct as many of these errors as we identified.

Reviewer #3 (BCR repertoire, epigenetic)(Remarks to the Author):

NCOMMS-18-18903

Mismatching at non-HLA loci, may influence the immune response of the recipient against the kidney donor graft and result in poor long-term graft outcomes. In this work, the authors have used high throughput BCR sequencing approach to analyze the BCR diversity and clonality in 27 kidney transplantation recipients, including non-progressors (NP; n=10), progressors with no rejection (PNR; n=10) and progressors with rejection (PR; n=7), before and 6 or 24 months after kidney engraftment. The authors conclude that individuals who developed rejection after kidney transplantation had a more diverse Ig VHDJH repertoire before the transplant than those who do not. Although it may be important to understand the impact of B-cell diversity on kidney transplant rejection, the putative differences in Ig VHDJH repertoire between NP and PNR or PR showed in the manuscript are likely resulted from experimental limitation.

1) The major conclusion of this manuscript is that individuals who developed rejection after kidney transplantation had a more diverse repertoire before the transplant. This is largely based on the results in Fig. 2 showing that the numbers of Ig VHDJH clones (with unique sequences) from NP samples were smaller than those from PNR or PR samples. However, as noticed by the authors, diversity measures may be affected by biological and technical sampling, and only a fraction of billions of B cells are represented. As indicated in Fig. S1, the lower numbers of

clones from the NP samples were correlated to a much fewer sequence reads generated from NP genomic DNA, as compared to PNR and, more significantly, PR. The numbers of sequence reads can be largely affected by several technical issues, such as PCR amplification to generate amplicons for miSeq sequencing, the amount and quality of genomic DNA, etc. A more accurate measurement would be the percentage of clones among the total reads from each individual sample. Indeed, as showed in Fig. 4 and Fig. S1, the proportion of clones among the total reads from PNR and PR samples are much lower than that from NP samples. At time 0, the VHDJH clones account of $789/1062=74.3\%$ of total reads from NP individual 1, while the VHDJH clones account of only $1581/4171=37.9\%$ and $4496/15495=29.0\%$ of total reads from PNR individual 20 and PR individual 25, respectively. This conclusion is the opposite to that made by the authors.

Thank you for your comment, this is one of the difficulties that B- and T-cell sequencing need to address. The ratio between number of clones and total number of reads cannot be used in this context due to the specificities of B- and T- cell sequencing. The total number of reads would measure the sequencing depth and the ratio will decrease when sequencing depth increases. The only way to use the total number of reads to normalize the number of clones would be if all samples had exactly the same total number of reads. In this type of studies, a measure of richness and/or diversity is normally used. In this context, the solution is to choose a diversity index that is independent of the sequencing depth. The richness measure has been widely used but may be biased and not completely independent on the sequencing depth, therefore diversity measures such as entropy are used. The different ways to measure diversity and the technical limitations when studying B- and T- cell sequencing have been widely discussed in the references 9, 11 and 12 of the paper. They all proposed to use different diversity index to account for the diversity of the immune repertoire. They also propose to use methods such as down-sampling in order to compare repertoires and avoid the technical issues and/or the sequencing depth problem. We have followed the pipelines proposed by the experts to make sure that our conclusion were not due to technical and/or sampling errors and in addition, we have also used Recon, which provides a robust estimates of the overall immune-repertoire. Based on the main analysis, the down-sampling strategy adopted and Recon estimates, we are pretty confident that our results are not biased due to technical or sampling errors.

2) In addition, the conclusion that “after 2 years of follow-up, patients who developed rejection showed an overall reduction of peripheral B-cell diversity” likely resulted from the reduced total reads from 24 PR samples (may be partially due to one less sample at this time point as compared to time 0, as indicated by Fig. S1) and increased total reads from time 24 NP samples. The increased total reads could be simply due to better PCR amplifications. Further the increase in diversity in NP across time after kidney transplantation, as emphasized in page 9, was likely also due to higher total reads from samples of time 6 and time 24 as compared to that of time 0.

The longitudinal analysis showed in figure 3 are performed using linear-mixed effect model where subject is used as random effect. The use of random effects offers several benefits when modeling longitudinal data. Among others, it provides a way to model correlation in unbalanced designs. The reason why unbalanced designs may be a problem is because when you have unbalanced groups, the F-statistic is not F-distributed. In this case, we have adjusted random effects with the lmer function in R and obtained the p-value using the likelihood ratio test that tests the difference in two nested models using the Chi square distribution. Therefore, the p-value showing the interaction between time

and clinical outcome and the decrease diversity in PR in comparison with the NP should not be an artifact due to unbalanced groups.

Regarding the problem with PCR amplifications, we agree that this is a big concern in this type of studies and that is why we have performed two different strategies, to correct for possible artifacts, in addition to the full data analysis. These two strategies (Recon and Down-sampling) provided similar results as shown in supplementary table S4. As suggested also by other reviewer, we have added more explanation regarding these two strategies for further clarification in lines 204-216.

Addition points:

1. Allograft rejection is associated with development of functional IgE, especially IgE specific for donor MHC antigens. It would make more sense to understand the diversification of IgE BCR.

This is a great point and we would like to focus future studies to look at IgE. Unfortunately, the current dataset doesn't allow us to carry out this type of analysis.

2. It would be important to know SHM levels in the three groups.

Thank you for bringing this to our attention - we looked to the SHM and performed the same type of analysis to see whether there was some difference between the three groups. We only found a trend for higher number of SHM for PR before Ktx, and trend for higher SHM in the IgD isotype in PR at 6 months post Ktx (p-value = 0.06). This was mentioned on lines 160-162 in the paper.

3. The amplicons used for sequences were amplified with AmpliTaq Gold, which is not a high fidelity polymerase and generates a high error-rate.

In the paper published by Beaulieu et al. (Nucl Acid Res 2001, PMID:11222761), the error rate of the AmpliTaq Gold is evaluated showing a 2.6×10^{-5} error rate, which should have a minimal effect on the identification of clonally-related sequences or estimation of somatic mutation rates in sequences. We have added a sentence with the reference in materials and methods.

4. The sequence of the PCR primers should be included.

The primers for the gDNA 454 set can be found in Boyd et al. 2009 [PMID: 20161664] which gives the gene-specific primers and the 454 sequence primers. In the case of the cDNA MiSeq data, the primers can be found in Roskin et al. [PMID 26311730]. We have added these two references to the materials and methods.

5. A Principal Component Analysis to determine whether the repertoires in each group are similar while different from other groups should be included.

This or any other multivariable analysis would be an excellent choice, but in this case the number of clones overlapping among patients is too small with only 23. Therefore, the data is too sparse for multivariable assessment with PCA.

Reviewer #4 (Biostat tech referee):

I'm confused about the diversity analysis. It seems there are 3 diversity analyses, 1) the full data, 2) the recon based diversity analysis and 3) the down-sampled strategy. Can you provide more details on the dis/agreement between the results from these methods?

We apologize for any potential confusion. Indeed, there are three diversity analyses, 1) the full data, 2) the down-sampling strategy and 3) the recon based. We have added a whole paragraph in the result section with more details about the disagreement between the results. You can find it on lines 204-216.

How was the FDR adjustment implemented in the IGHV analysis? E.g., was it Benjamini and Hochberg?

Yes, we apologize for not having these details clearly stated. Benjamini and Hochberg was applied, and we added this information to line 658 in materials and methods.

Reviewers' comments:

Reviewer #1 (Remarks to the Author):

Most of my concerns have been addressed. However, several of the findings are quite soft (of marginal statistical significance or failing to achieve significance) and the authors should be careful not to overinterpret their results and not to speculate too extensively about them.

Specific comments:

The meaning of the sentence in the abstract stating that "specific peripheral IGHV gene overexpression may imply a B-cell activation of the immune repertoire during acute rejection" is unclear. If the authors are trying to say that expansion of clones using specific IGHV genes in multiple rejecting patients may implicate particular common antigens in driving rejection, they should say so. The current sentence is vague and nonsensical.

The highly speculative discussion of the implications of the diversity data on p.21, with mention of "huge implications" should be shortened and toned down, as there are no data to indicate that diversity has predictive value for an individual patient and could be used as predictive biomarker. There is considerable overlap between patients in different groups and the data are far from supporting use of this parameter as a biomarker.

Line 385 states that the study "highlights the importance of clonotypes and IGHV genes in the process of rejection". This statement is inaccurate. The study merely shows an association between expansions of some clonotypes and IGHV genes and rejection. No causal relationship is demonstrated.

The new speculation in lines 423-426 should be removed or shortened.

Overall, the manuscript is poorly written. The authors should go through the manuscript in detail to correct the numerous errors in language use and tighten up the writing.

Reviewer #2 (Remarks to the Author):

The revised paper remains limited by the size of the original clinical study and thus unfortunately cannot be further enhanced in terms of subject numbers.

The clinical aspects of the questions put to the authors have been answered in the authors' letter but have not all been explained in the manuscript.

eg The time to rejection has been described in the letter and placed in the table but needs to have the range of time from transplant to rejection added to make clear that all rejections were after 6 months.

The remaining clinical concern is"

Late rejection (ie after 6 months) usually has a very serious and different impact to early post transplant rejection. It is usually associated with development of DSA and carries a poor outcome compared to steroid treated and responsive early rejection. I find it curious that the rejections were all late and were relatively benign and not associated with DSA development. Can the authors provide

the readers more assurance and explanation on this issue based on national experience with these protocols or other perspectives?

The word 'huge' used in the last sentence of the conclusions is at odds with the more temperate language used throughout the rest of the paper. It could be deleted without losing the message.

Reviewer #3 (Remarks to the Author):

The authors have dealt satisfactorily with my original criticisms, and edited the revised manuscript accordingly.

Nevertheless, the significance of this work, even in its revised form, remains modest, not up to the level required by Nature Communications. This manuscript would be more appropriate for a specialistic journal, such as "Kidney".

Reviewer #4 (Remarks to the Author):

Thank you for addressing my previous concerns. Your answers were satisfactory.

My other concern is in the clone specific analysis and multiple testing. For that analysis, it's stated that 8,4, and 21 markers were close but did not meet a multiple testing adjustment. It seems to suggest that "close" means a p-value of 0.05 which is not close to significant when multiple testing is considered. To me, when nothing is significant after multiple testing that should end the conversation, but those markers are detailed in Table 2 as if they were significant. To me, if a clone does not survive a reasonable multiple testing adjustment then it's not worthy of being discussed. To do so, creates the risk of that marker becoming significant because "it was established in previous literature."

We appreciate the valuable comments from the editors and reviewers and believe we have addressed all of the reviewers' concerns in the new manuscript. We have toned down and clarified the text requested by the reviewers and revised the title/abstract of the manuscript accordingly. We have also read carefully the manuscript correcting all the grammatical errors. We have tracked the changes in the main text. Please see our point-to-point responses in bold below.

Reviewers' comments:

Reviewer #1 (Remarks to the Author):

Most of my concerns have been addressed. However, several of the findings are quite soft (of marginal statistical significance or failing to achieve significance) and the authors should be careful not to overinterpret their results and not to speculate too extensively about them.

We appreciate the concern. We agree that some findings are softer than others, but we also believe that this is the first analysis done in the study of B cell repertoire in kidney transplant and that these results will enable additional analysis on this topic. Nevertheless, we have toned down the paper and are very careful with the speculation to avoid any overinterpretation of the findings.

Specific comments:

The meaning of the sentence in the abstract stating that “specific peripheral IGHV gene overexpression may imply a B-cell activation of the immune repertoire during acute rejection” is unclear. If the authors are trying to say that expansion of clones using specific IGHV genes in multiple rejecting patients may implicate particular common antigens in driving rejection, they should say so. The current sentence is vague and nonsensical.

Thank you for this clarification. One of the points that was suggested by the reviewer in the previous review was to explore whether there was an overrepresentation of the IGHV3-23 and precisely that was what we found, so the proposed sentence is very relevant and clarifies the findings. We have modified this in the abstract and also in the discussion to clarify the concept of gene usage.

The highly speculative discussion of the implications of the diversity data on p.21, with mention of “huge implications” should be shortened and toned down, as there are no data to indicate that diversity has predictive value for an individual patient and could be used as predictive biomarker. There is considerable overlap between patients in different groups and the data are far from supporting use of this parameter as a biomarker.

Thank you for the comment. We are very excited about these findings and that is probably reflected in the text. We have toned down this speculation and modified the sentence in the discussion to avoid any overinterpretation. The sentence now is: “The next step is to demonstrate the predictive value of B-cell repertoire diversity, providing potential better biomarkers for prediction of rejection before engraftment, and the possibility of being implemented in clinical care and immunosuppression choices before and after Ktx.”

Line 385 states that the study “highlights the importance of clonotypes and IGHV genes in the process of rejection”. This statement is inaccurate. The study merely shows an association between expansions of some clonotypes and IGHV genes and rejection. No causal relationship is demonstrated.

Correct, there is no casual relationship. We have modified the sentence stating just that there is an association and the causal link should be explored in future studies.

The new speculation in lines 423-426 should be removed or shortened.

This has been removed.

Overall, the manuscript is poorly written. The authors should go through the manuscript in detail to correct the numerous errors in language use and tighten up the writing.

The manuscript has been edited extensively to correct the errors.

Reviewer #2 (Remarks to the Author):

The revised paper remains limited by the size of the original clinical study and thus unfortunately cannot be further enhanced in terms of subject numbers.

Yes, we cannot extend the subject numbers in this study, but we believe this is the first study on this topic and this paper sets the basis for future directions.

The clinical aspects of the questions put to the authors have been answered in the authors' letter but have not all been explained in the manuscript.

eg The time to rejection has been described in the letter and placed in the table but needs to have the range of time from transplant to rejection added to make clear that all rejections were after 6 months.

In the last review, we added the time to rejection in the Table 1 in mean and SD format, but to clarify that all rejections were after 6 months, we have modified the format to mean [min-max] and added a sentence in text so it is clear that all the rejections were after 6 months. This was also commented previously in the discussion section. In addition, to facilitate the interpretation of the analysis we are publishing all the raw data for Table 1 together with the sequencing data under the accession number specified at the end of the paper, so people will have the opportunity to find the individual numbers per sample.

The remaining clinical concern is"

Late rejection (ie after 6 months) usually has a very serious and different impact to early post transplant rejection. It is usually associated with development of DSA and carries a poor outcome compared to steroid treated and responsive early rejection. I find it curious that the rejections were all late and were relatively benign and not associated with DSA development. Can the authors provide the readers more assurance and explanation on this issue based on national experience with these protocols or other perspectives?

The parent study that these patients were enrolled from had an overall low rate (17%) of biopsy confirmed acute rejection (mean [min – max] = 12 [6 – 24] months rejection time). These patients were all at low immunologic risk for rejection (peak panel reactive antibody sensitization status <20%), and also had low rates of generation of donor specific antibody (DSA) and in fact only 2 of the rejection phenotype patients included in the analysis had DSA. The generation of DSA to HLA, and MICA were measured in all serial sera over the course of the study (Chaudhuri et al, JASN, 2013) and have been reported previously. National experience with similar immunosuppressive protocols in similar patient cohorts have confirmed similar good clinical outcomes and low rejection rates (doi: 10.1038/ki.2009.248; PMID:16968478; PMID:18637457; doi: 10.1155/2014/171898). The late rejections that can be more aggressive and more likely to be DSA positive, and often driven by non-adherence with medications, are not observed in this study, likely due to the study effect of very close clinical oversight. We have added this paragraphed to the text as an explanation.

The word 'huge' used in the last sentence of the conclusions is at odds with the more temperate language used throughout the rest of the paper. It could be deleted without losing the message.

We have deleted the word “huge” and also, as suggested by other reviewers, we have toned down the whole paper.

Reviewer #4 (Remarks to the Author):

Thank you for addressing my previous concerns. Your answers were satisfactory.

We are glad we were able to address the reviewer’s comments.

My other concern is in the clone specific analysis and multiple testing. For that analysis, it's stated that 8,4, and 21 markers were close but did not meet a multiple testing adjustment. It seems to suggest that "close" means a p-value of 0.05 which is not close to significant when multiple testing is considered. To me, when nothing is significant after multiple testing that should end the conversation, but those markers are detailed in Table 2 as if they were significant. To me, if a clone does not survive a reasonable multiple testing adjustment then it's not worthy of being discussed. To do so, creates the risk of that marker becoming significant because "it was established in previous literature."

We completely agree with your comment and the multiple testing (MT) correction threshold. Nevertheless, MT correction is also an arbitrary threshold and as we suggest in this commentary (<https://bit.ly/2svpA4x>) p-values are not the best way to declare that a results is relevant; having said that, we have changed the wording in the results section to specify in the text that these clones do not pass MT correction and are nominally significant (p-value < 0.05) to avoid the risk of the marker becoming significant because "it was established in previous literature.” In addition, to avoid any misinterpretation of the results, we have removed the sentence in the discussion section about this specific result.

REVIEWERS' COMMENTS:

Reviewer #2 (Remarks to the Author):

The tone has improved generally throughout the paper, as requested by more than one reviewer.

The clinical explanation of the late rejections being relatively benign because of close observation is really just speculation and perhaps best left out, just leaving the facts for the reader to consider.

I have no other issue to raise.

Reviewer #4 (Remarks to the Author):

My previous concern was regarding the clonal analysis. Specifically including a table of markers that were only nominally significant ($p < 0.05$) rather than being significant after adjusting for multiple testing.

Part of the response was to cite another of their manuscripts which advocated for other modeling approaches rather than p-values to establish significance. To me, this response is not relevant as they do not employ other alternative modeling approaches to establish significance. The other part of the response is to remove a sentence in the discussion about these results however the table of results (table 2) is still retained in the manuscript.

We appreciate the valuable comments from the reviewers and believe we have addressed all of the reviewers' concerns in the new manuscript. Please see our point-to-point responses in bold below.

REVIEWERS' COMMENTS:

Reviewer #2 (Remarks to the Author):

The tone has improved generally throughout the paper, as requested by more than one reviewer.

Thanks.

The clinical explanation of the late rejections being relatively benign because of close observation is really just speculation and perhaps best left out, just leaving the facts for the reader to consider.

We have deleted this sentence. Thanks for the advice.

I have no other issue to raise.

Reviewer #4 (Remarks to the Author):

My previous concern was regarding the clonal analysis. Specifically including a table of markers that were only nominally significant ($p < 0.05$) rather than being significant after adjusting for multiple testing.

Part of the response was to cite another of their manuscripts which advocated for other modeling approaches rather than p-values to establish significance. To me, this response is not relevant as they do not employ other alternative modeling approaches to establish significance. The other part of the response is to remove a sentence in the discussion about these results however the table of results (table 2) is still retained in the manuscript.

We are sorry if the manuscript we cited did not answer the reviewer concern. Our intention was to show that there is a discussion in the field regarding the use of p-values agreeing with the reviewer concern. To make sure that there is no misunderstanding with this issue, we have moved the table to supplementary material.